# C₃N nanodots inhibits Aβ peptides aggregation pathogenic path in Alzheimer's disease

Xiuhua Yin[1,2,6], Hong Zhou[1,6], Mengling Zhang[2,3,6], Juan Su[4], Xiao Wang[2], Sijie Li[4], Zaixing Yang [1,4] ✉, Zhenhui Kang [2,3] ✉ & Ruhong Zhou [1,5] ✉

Despite the accumulating evidence linking the development of Alzheimer's disease (AD) to the aggregation of Aβ peptides and the emergence of Aβ oligomers, the FDA has approved very few anti-aggregation-based therapies over the past several decades. Here, we report the discovery of an Aβ peptide aggregation inhibitor: an ultra-small nanodot called C₃N. C₃N nanodots alleviate aggregation-induced neuron cytotoxicity, rescue neuronal death, and prevent neurite damage in vitro. Importantly, they reduce the global cerebral Aβ peptides levels, particularly in fibrillar amyloid plaques, and restore synaptic loss in AD mice. Consequently, these C₃N nanodots significantly ameliorate behavioral deficits of APP/PS1 double transgenic male AD mice. Moreover, analysis of critical tissues (e.g., heart, liver, spleen, lung, and kidney) display no obvious pathological damage, suggesting C₃N nanodots are biologically safe. Finally, molecular dynamics simulations also reveal the inhibitory mechanisms of C₃N nanodots in Aβ peptides aggregation and its potential application against AD.

Alois Alzheimer reported the first case of Alzheimer's disease (AD) in 1906[1,2]. Now, more than one century later, AD remains an unresolved public health problem worldwide[3]. AD is a progressive neurodegenerative disease associated with insidious onset and slow progression of behavioral and cognitive dysfunction. The severity of the AD from early stage[4] advances to obvious symptoms which further aggravates the need to utilize immediate remedies against the progression of the disease. Moreover, the incidence of AD also increases with the increasing age reflected by the increasing rate of ~27.6% in 65–74 year-old people to ~36.4% in people over 80 years old[5]. This significant increase with age also poses a worldwide threat of acquiring AD among elderly population. This also urges the need of developing novel and effective AD management therapies for clinical purposes.

Growing evidence suggests the aggregation of Aβ peptides is highly related with synaptic dysfunction, neuroinflammation, oxidative stress damage, neurotoxicity mediated by the triggered hyperphosphorylation of downstream Tau protein, as well as the ultimate cell death[6–9]. Additionally, Aβ oligomers also drive pathology by damaging cell membranes, activating receptors, disrupting signaling, impairing mitochondria, perturbing the trans-Golgi network, inducing endoplasmic reticulum stress, causing endosomes/lysosomal leakage, and triggering macroautophagy[6,10–15]. In contrast, reversal of the Aβ peptides aggregation process also offers a suitable therapeutic strategy against AD. However, the successful implementation of this concept remains a huge challenge despite decades of effort along this direction. On the other hand, the lack of effective drugs against AD,

[1]Institute of Quantitative Biology, Shanghai Institute for Advanced Study, College of Life Sciences, Zhejiang University, Hangzhou 310027, China. [2]Jiangsu Key Laboratory for Carbon-based Functional Materials and Devices, Institute of Functional Nano and Soft Materials (FUNSOM), Soochow University, Suzhou 215123, China. [3]Macao Institute of Materials Science and Engineering (MIMSE), MUST–SUDA Joint Research Center for Advanced Functional Materials, Macau University of Science and Technology, Taipa 999078 Macao, China. [4]State Key Laboratory of Radiation Medicine and Protection, School for Radiological and Interdisciplinary Sciences (RAD-X), Soochow University, Suzhou 215123, China. [5]Department of Chemistry, Columbia University, New York, NY 10027, USA. [6]These authors contributed equally: Xiuhua Yin, Hong Zhou, Mengling Zhang. ✉e-mail: zxyang@suda.edu.cn; zhkang@suda.edu.cn; rhzhou@zju.edu.cn

with only two FDA-approved options available, such as aducanumab[16] and lecanemab[17], still raising high demand for alternate therapeutic options. Encouragingly, in a phase-III clinical trial, another monoclonal antibody agent called donanemab exhibited promising positive results[18]. Besides, other anti-AD agents (including peptides[19,20], polymers[21,22], small drug molecules[23–26], and metal oxides[27]) show only a very mild inhibition effect on Aβ peptides aggregation. Recently, nanomaterials (NMs) (e.g., graphene oxide[28], fullerenes[29,30], quantum dots[31], carbon nanotube[32], and g-C$_3$N$_4$[33,34]) have been reported to inhibit, directly or indirectly, the aggregation of Aβ peptides, including both the inhibition of oligomer fibrillization and disaggregation of mature fiber in vitro. The potential of these NMs to inhibit aggregation is closely related to their physical and chemical properties, including size, curvature, and modifications[35,36]. But very few of them can still work in vivo. Interestingly, graphene quantum dots were also found to inhibit α-synuclein aggregation, disassociate mature fibrils, and penetrate the blood-brain barrier (BBB) leading to ultimate protection of dopamine neurons[37]. Therefore, the use of nanomaterials may offer valuable alternate source as therapeutic agents for protein conformational diseases (e.g., AD, Parkinson's disease, Huntington's disease, Type 2 diabetes).

In this study, we demonstrate that C$_3$N nanodots can significantly inhibit Aβ peptides aggregation and disaggregate mature Aβ fibrils, relieve aggregation-induced neuron cytotoxicity, rescue neuronal death, protect neurites from damage, and exhibit only mild cytotoxicity both in vitro and in vivo. Moreover, the intraperitoneal administration of C$_3$N nanodots for 6 months significantly improves the learning and spatial memory abilities of APP/PS1 in double transgenic AD mice. Additionally, the underlying molecular mechanism of Aβ peptide aggregation inhibition by C$_3$N nanodots has also been explored using all-atom molecular dynamics (MD) simulations. Thus, we believe our current study provides deep insights into the anti-Aβ peptides aggregation capability of C$_3$N nanodots and its potential application against AD.

## Results

### C$_3$N nanodots inhibit Aβ$_{42}$ peptides fibrillization in vitro

C$_3$N nanodots were synthesized by polymerization of 2,3-diaminophenazine using hydrothermal synthesis following a previous report[38]. The synthesized nanodots had an average lateral size of 4.5 ± 0.4 nm (Fig. 1a) with a lattice spacing of 0.21 nm, which corresponds to the (100) plane of graphite. Meanwhile, these nanodots had a height of less than 1 nm, indicating a stacking arrangement of one or two layers (Supplementary Fig. 1). Initially, the identification and characterization of C$_3$N nanodots were performed using several spectroscopic techniques including UV–visible (UV–Vis) absorption spectroscopy, Fourier transform infrared (FTIR) spectroscopy and X-ray photoelectron spectroscopy (XPS). (Supplementary Fig. 2).

We first studied the role of C$_3$N nanodots towards the aggregation behavior of Aβ$_{42}$ peptides, which were shown to have more implications than Aβ$_{40}$ in forming neurotoxic assemblies and causing AD pathogenesis[39,40]. In the absence of C$_3$N nanodots, Aβ$_{42}$ peptides aggregated into mature amyloid fibers, as demonstrated by various experimental procedures. This included the utility of ThT fluorescence, dot blot assay, atomic force microscope (AFM), transmission electron microscope (TEM), and CD spectroscopy. During these investigations, C$_3$N nanodots effectively inhibited the aggregation of Aβ$_{42}$ peptides (Fig. 1). It was evident from delayed aggregation kinetics and reduced ThT fluorescence intensity (after convergence of aggregation process) following C$_3$N nanodots treatment. The inhibition strength was found positively correlated with C$_3$N nanodots treatment concentration (Fig. 1b). It should be noted that, under the concentrations examined, C$_3$N nanodots did not entirely inhibit the aggregation of peptides. The final peptide self-assembly samples were also examined through dot blotting using an amyloid fiber conformation-specific

antibody (mOC87)[41]. Notably, amyloid fiber content decreased with the increasing concentration of C$_3$N nanodots during treatment (Fig. 1c). This confirmed the inhibition function of C$_3$N nanodots against peptides aggregation. Morphologically, Aβ$_{42}$ peptides aggregated to long and well-defined mature fibers after 24 h in PBS without C$_3$N nanodots, as demonstrated through AFM and TEM imaging (Fig. 1d and Supplementary Fig. 3). In contrast, incubation with C$_3$N nanodots for 24 h resulted in a gradual morphologic change of Aβ$_{42}$ peptides self-assembly samples from long mature fibers to diffused punctiform structures. Furthermore, it is worth noting that the aggregation of N-truncated Aβ peptides (AβpE3) and Aβ$_{40}$ is also likely to contribute to the molecular pathology of AD. Our investigation also delved into the impact of C$_3$N nanodots on the aggregation of these two peptide species. Remarkably, comprehensive evaluations encompassing ThT fluorescence, dot blot assay, CD spectra, TEM, and AFM unequivocally demonstrated the potent ability of C$_3$N nanodots to effectively impede the aggregation of these peptides (Supplementary Figs. 4 and 5).

To our surprise, C$_3$N nanodots exhibited an exceptional capacity to disassemble mature fibrils of Aβ$_{42}$ as well. The convergence of evidence from ThT fluorescence, dot blot assay, CD spectra, AFM, and end-to-end distance results collectively substantiate that, in a concentration and duration-dependent manner, the co-incubation of mature fibrils with C$_3$N nanodots led to the gradual dismantling of these originally long, well-defined fibrils into smaller, amorphous entities (Supplementary Fig. 6). Overall, these results suggested that C$_3$N nanodots effectively reverse the aggregation of Aβ peptides.

To further unveil the regulating process and underlying molecular mechanisms of C$_3$N nanodots towards inhibiting aggregation of these peptides, we then performed all-atom molecular dynamics (MD) simulations (Supplementary Fig. 7). In the absence of C$_3$N, two Aβ$_{42}$ peptides self-assembled into a partially ordered structure (containing β-sheets). However, C$_3$N nanodot application significantly inhibited the formation of any β-sheets. For instance, in two out of three trajectories (run 1 and run 3), very rare β-sheet contents were formed (i.e., in run 2, β-sheet appeared at t = 80 ns, then disappeared at t = 340 ns) (Fig. 1e and Supplementary Fig. 8). Convergence of the simulations (>900 ns) demonstrated an overall decrease in the β-sheet content of ~10.6 ± 1.5% without C$_3$N to 0.2 ± 0.6% with C$_3$N. Simultaneously, the random-coiled and bend components increased from ~37.0 ± 2.4% to ~40.6 ± 1.7%, and ~13.5% to ~20.3% (Fig. 1f), respectively. These findings suggested that C$_3$N nanodot effectively redirects Aβ$_{42}$ peptides self-assembly to disordered structures. Moreover, CD spectroscopy confirmed that C$_3$N nanodots redirected the secondary structure of Aβ$_{42}$ peptides (at time = 24 h) from the β-sheet-rich to disordered random-coiled conformations (Fig. 1g). These results sufficiently demonstrate the structural modulating role of C$_3$N nanodot in impeding the aggregation of Aβ$_{42}$ peptides.

The detailed interaction energies including both van der Waals (vdW) and electrostatic (elec) interactions between C$_3$N and peptides were also explored (Fig. 1h). This was performed by analyzing the key binding configurations in a typical trajectory to better illustrate the binding mechanisms. Driven by vdW and hydrophobic interactions, one peptide was adsorbed onto the surface of C$_3$N (time = 1 ns) and strengthened by π–π stacking interactions (F4 and F20) (time = 10 ns). At time = 13 ns, another peptide was adsorbed onto the edge of C$_3$N by electrostatic attractions between E11, D7, and E3 residues with –NH$_3^+$ groups at the edge of C$_3$N nanodot. At time = 33 ns, this peptide was fully adsorbed onto the other side of C$_3$N nanodot via vdW and π–π stacking interactions. After 96 ns, the adsorption process converged. At this state, most hydrophobic and aromatic residues were adsorbed onto the C$_3$N nanodot surface. Meanwhile, some charged or polar residues formed salt-bridge or hydrogen bonds with edge groups (e.g., –COO$^-$ and –NH$_3^+$) of C$_3$N nanodot while suppressing subsequent aggregation of peptides. Hence, the strong adsorption between

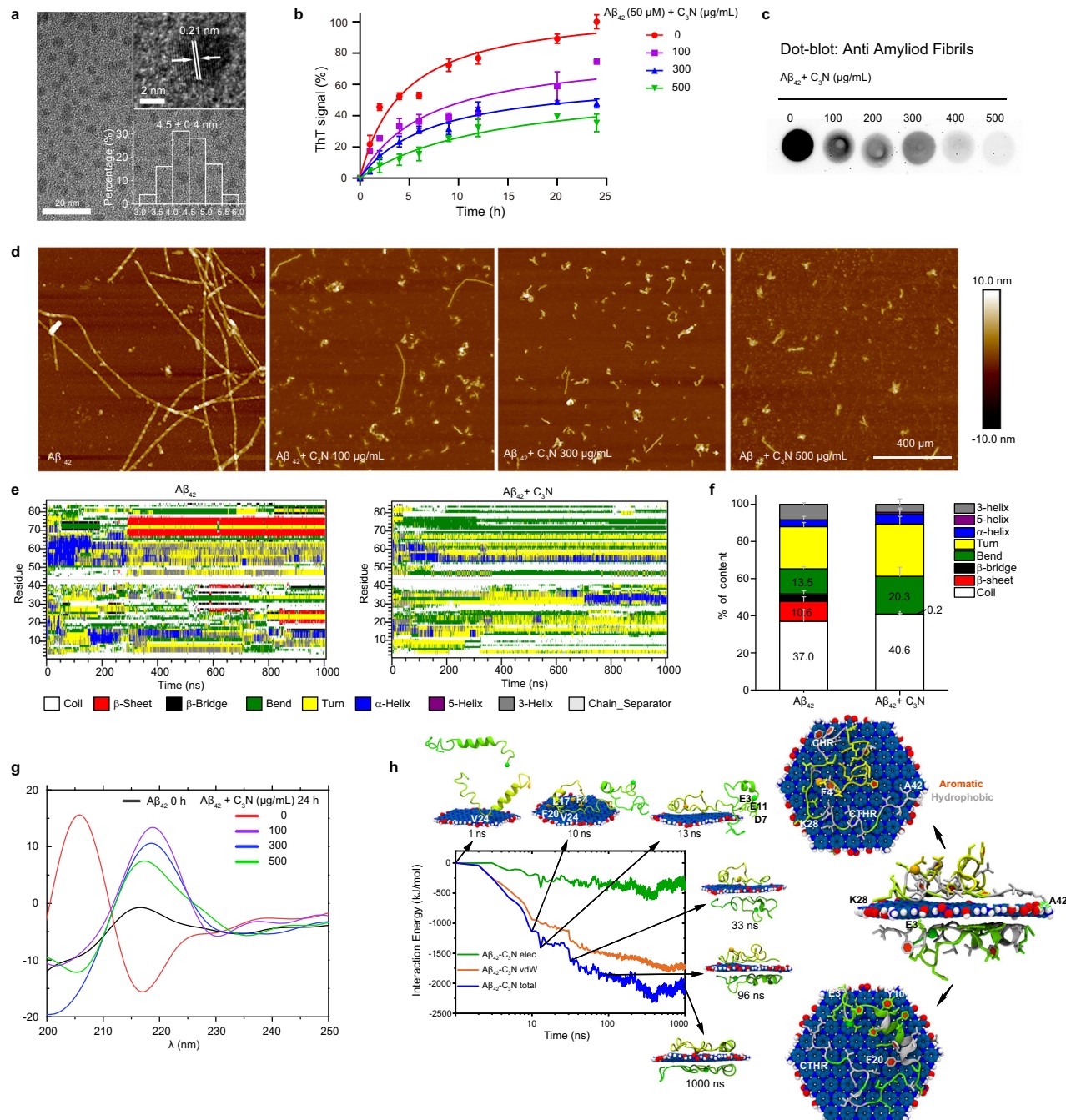

**Fig. 1 | C₃N nanodots inhibit Aβ₄₂ fibrillization in vitro. a** Transmission electron microscopy (TEM) image, crystal structure (top right corner, HRTEM image), and lateral size distribution (bottom right corner, histogram) of C₃N nanodots. The image is representative of three independent experiments. **b** The influence of C₃N nanodots on Aβ₄₂ peptides (50 µM) aggregation was detected by ThT fluorescence. Data are presented as mean ± SD, $n = 3$ biological replicates and signals were normalized by setting the maximal ThT signals to 100%. **c** The formation levels of amyloid fiber under different conditions were detected by dot blot assay using Aβ fibrils conformation specific antibody (mOC87), at time = 24 h. Immunoblots are from one experiment representative of three independent experiments with similar results. **d** Representative AFM images of Aβ peptides untreated/treated with C₃N

nanodots (0, 100, 300, and 500 µg/mL) for 24 h. $n = 3$ independent experiments. **e** Time evolutions of the secondary structure of each residue in two Aβ₄₂ peptides. The secondary structures of residues were assigned using the DSSP definition[72]. **f** The proportions of each structural component in the peptides. **g** CD spectra of Aβ peptides at 0 and 24 h in the absence of C₃N nanodots and after incubation with C₃N nanodots for 24 h. **h** The nonbonded interaction energies (including electrostatic (elec), van der Waals (vdW) interactions, and a total of them) between C₃N nanodots and peptides and key binding configurations during the process. Green dashed lines indicate hydrogen bonds, and the hydrophobic and hydrophilic (polar/charged) residues are shown with silver and green, respectively. Source data are provided as a Source data file.

peptides and C₃N nanodot was collectively driven by a combination of vdW and electrostatic, hydrophobic, hydrogen bonding, and π–π stacking interactions, with the vdW interaction dominating (Fig. 1h), to induce disruption in peptides self-assembly and form disordered structures.

Moreover, we conducted a comparative analysis of the inhibitory effects of stacked C₃N nanodots (two layers), nano graphite (GRA) (simulated by two layers of stacked graphene), and fullerene (e.g., C₆₀) on the aggregation of Aβ₄₂ peptides. The findings clearly indicate that C₃N nanodots exhibit a relatively stronger capability in inhibiting Aβ

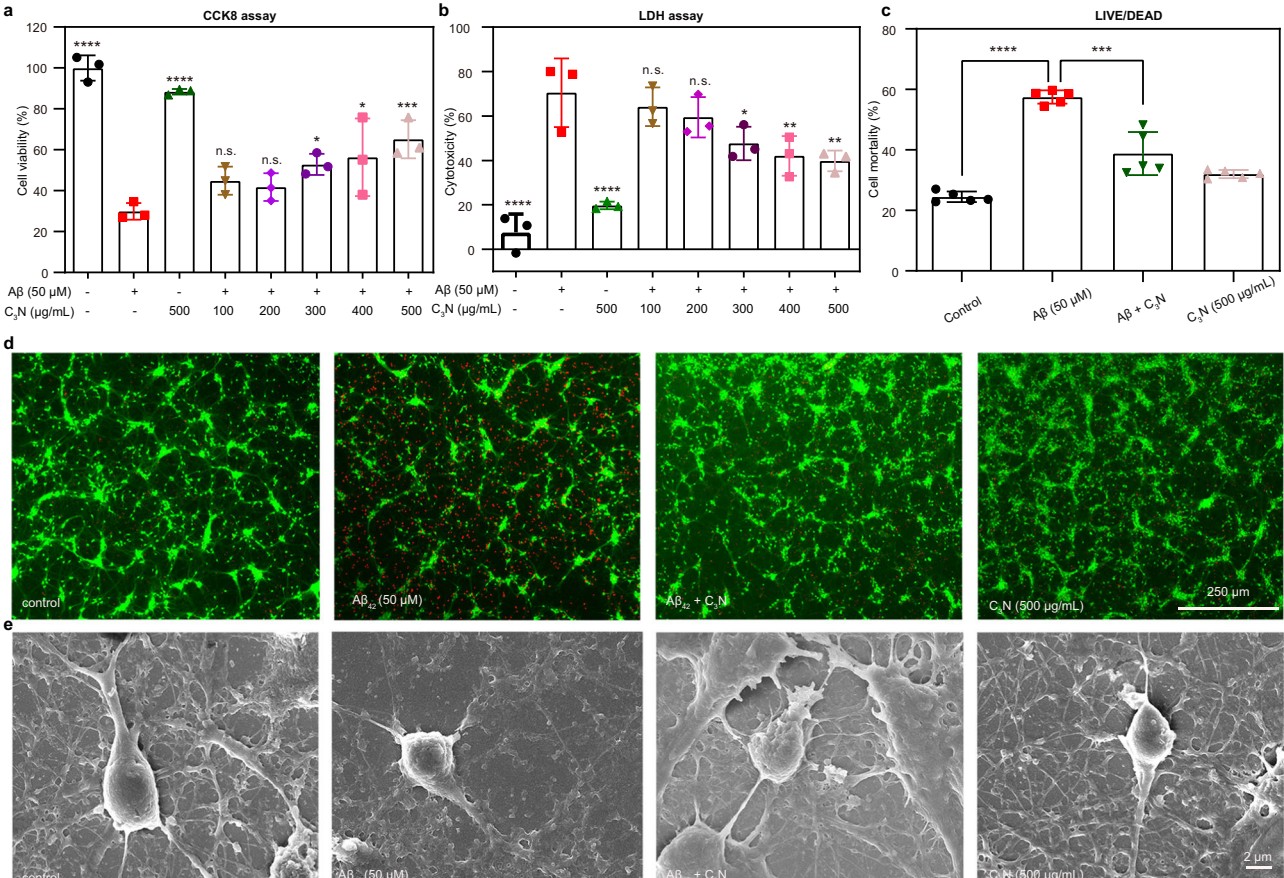

**Fig. 2 | C₃N nanodots reduce Aβ₄₂-aggregation-induced cytotoxicity.** Neurons were cultured with/without Aβ₄₂ peptides for 24 h. Cytotoxicity of Aβ₄₂ aggregates in the presence/absence of different concentrations of C₃N nanodots for 24 h to primary neurons was assayed by **a** CCK8 ($P < 0.0001$, $P < 0.0001$, $P = 0.0300$, 0.0114, and 0.0009, respectively) and **b** LDH-release ($P < 0.0001$, $P < 0.0001$, $P = 0.0312$, 0.0065, and 0.0035, respectively), $n = 3$ independent experiments. Statistical significance was determined by one-way ANOVA in (**a–c**) with $p < 0.05$ considered statistically significant. **c, d** Live/dead staining experiments to examine whether C₃N nanodots alleviate the cytotoxicity of neurons induced by Aβ₄₂

peptides. $n = 5$ independent experiments. Statistical significance was determined by unpaired Student's t test (two-tailed) with $P < 0.05$ considered statistically significant ($P < 0.0001$, $P = 0.0005$). **d** Photomicrographs of live/dead assay showing live (green cell body) and dead (red nuclei) cells in each group. **e** Morphology of cells in each group was observed under SEM. The images are from one experiment representative of three independent experiments with similar results. All data are presented as mean ± SD. *$P < 0.05$, **$P < 0.01$, ***$P < 0.001$, and ***$P < 0.0001$ vs 50 μM Aβ₄₂ group. n.s. = not significant. Source data are provided as a Source data file.

peptide aggregation compared to the other two alternatives (Supplementary Fig. 9). Notably, the electrostatic potential (ESP) calculations of C₃N nanodots reveal the presence of numerous polar C–N bonds and charged edge groups (e.g., –COO⁻ and –NH₃⁺), resulting in a significantly polar surface for C₃N nanodots (Supplementary Fig. 10), which significantly differs from GRA and fullerene represented by Lennard–Jones (LJ) particles. These distinct surface properties of C₃N nanodots enable more effective suppression of peptide aggregation through multiple interactions, including vdW, electrostatic, hydrophobic, hydrogen bonding, and π–π stacking interactions. Furthermore, these surface properties confer advantages upon C₃N nanodots, such as superior water dispersity and compatibility with cell membranes, in contrast to highly hydrophobic candidates like nano GRA, fullerene, and others.

## C₃N-nanodots alleviate neuron cytotoxicity induced by Aβ₄₂ peptides and demonstrate superior cytocompatibility

As shown above, C₃N nanodots exhibited an effective inhibiting function against Aβ₄₂ peptides aggregation at molecular level. At this stage, it was logical to examine whether C₃N nanodots alleviate aggregation-induced neuron cytotoxicity (Fig. 2). Herein, we analyzed primary neuron cells viability and toxicity under different conditions using cell counting kit 8 (CCK-8), Lactate Dehydrogenase

(LDH), and Live/Dead assays. The CCK-8 assay results demonstrated that Aβ₄₂ peptides aggregation causes severe toxicity in neurons. This was found after neuronal cells incubation with 50 μM Aβ₄₂ peptides for 24 h which resulted in a survival rate of only ~29.89 ± 3.98%. However, increased treatment concentration with C₃N nanodots resulted in improved cell survival rate: ~44.83 ± 6.90% (100 μg/ml) to ~65.52 ± 9.12% (500 μg/mL) (Fig. 2a). Hence, C₃N nanodots dose-dependently relieved Aβ₄₂ peptides aggregation-induced neuron cytotoxicity, which was further confirmed by LDH (Fig. 2b) and Live/Dead experimental (Fig. 2c, d) results. In addition, the cytotoxicity of C₃N nanodots was found very mild with C₃N nanodots administered at 500 μg/mL resulting a neuronal survival rate of ~88.47 ± 1.36%. We further investigated the morphologies of neurons under different conditions using scanning electron microscope (SEM) technology (Fig. 2e). Normal neurons presented in a plump-pear shape with many dendrites. However, Aβ₄₂ aggregation-induced significant deformations of neurons, e.g., the cellular body shrunk notably and was accompanied by severe dendrites loss. In contrast, the treatment with C₃N nanodots resulted in well maintained dense dendrites suggesting inverse effect against the toxicity caused by Aβ₄₂ peptides aggregation in neurons. It also distinguished the mild influence of C₃N nanodots on the shape of neurons. These effects predominantly stem from the fact that C₃N nanodots facilitate

the reversal of Aβ$_{42}$ peptide aggregation. Additionally, the adsorption of peptides onto the surface of C$_3$N nanodots, leading to a decrease in peptide concentration in the solution, is expected to play a role in alleviating the cytotoxicity of Aβ$_{42}$ peptides to neurons. It is noteworthy that both the drug and peptide concentrations employed at the cellular level are relatively high. As we transition to the animal level, attaining elevated drug concentrations requires surmounting the BBB, a challenge that could potentially be addressed through sustained and long-term administration strategies.

In addition, the cytotoxicity of C$_3$N nanodots in several cell lines was also examined, including red blood cells (RBCs), primary mouse neuron (Neuron), rat adrenal chromaffin cell tumor cells (PC12), primary rat astrocyte (Astrocyte), human umbilical vein endothelial cells (HUVECs), human neuroblastoma cells (sh-sy5y). The results showed that C$_3$N nanodots possess decent cytocompatibility among all tested cell lines (Supplementary Fig. 11 and Supplementary Fig. 12). Moreover, C$_3$N nanodots showed much superior biocompatibility than GO nanosheets (Supplementary Fig. 13). This revealed that C$_3$N nanodots alleviate neuron cytotoxicity, reduce cell death, protect Aβ$_{42}$ aggregation-induced axonal and dendritic damages and demonstrate remarkable cytocompatibility.

## C$_3$N-nanodots improve the learning and spatial memory capabilities of APP/PS1 mice with limited biotoxicity

Following the encouraging in vitro findings, we sought to determine whether C$_3$N nanodots have neuroprotective functions towards AD mice via inhibition of Aβ peptides aggregation. For this purpose, we used APP/PS1 double transgenic mice as the model AD organism. Here, male mice were chosen exclusively for this study as they may have a relatively stable hormone level and much less estrogen's impact[42–44]. It thus allows for a more accurate observation and evaluation of disease progression and pathological changes upon the application of nanomedicine. This in-vivo model overexpresses Aβ peptides in the brain by inducing amyloid plaque formation which eventually leads to the occurrence of AD symptoms[45,46]. The expression of Aβ peptides in APP/PS1 mice begins at 3–4 months of age. Thus, we treated the APP/PS1 mice with C$_3$N nanodots-saline solution per day from 3 to 9 months via intraperitoneal injection. APP/PS1 mice received saline only were set as the positive control group, and wild-type (WT) mice with non-intervention were set as the negative control. After six months of C$_3$N nanodots injection vs. no injection, the cognitive function of APP/PS1 mice were examined using the Morris water maze and novel object recognition tests (Fig. 3).

In order to further ascertain the ability of C$_3$N nanodots to traverse the BBB and accumulate within the brain, an essential prerequisite for their potential application in AD treatment, we employed Cy5.5-modified C$_3$N nanodots (referred to as C$_3$N-Cy5.5) to enable fluorescence imaging. For this purpose, we conducted experiments utilizing healthy C57BL/6J mice, which were divided into five distinct groups: (1) a control group without any treatment, and (2) groups that received intraperitoneal (i.p.) administration of C$_3$N-Cy5.5 nanodots for 8 h, (3) 24 h, (4) 48 h, and (5) 1 week. The administration of C$_3$N-Cy5.5 nanodots was accomplished via injections of a PBS solution at a relatively high dosage of 200 mg/kg ($n = 3$ per group) to ensure optimal imaging. As shown in Fig. 3a, b, at time = 8 h, the fluorescence emanating from C$_3$N-Cy5.5 nanodots within the brain was discernible. Subsequently, the highest intensity of fluorescence was observed at 48 h post-injection, gradually diminishing to undetectable levels after one week. These compelling outcomes substantiate the remarkable capacity of C$_3$N nanodots to successfully penetrate the BBB, thus establishing a crucial foundation for their potential therapeutic application in AD.

Then, we refined the optimal C$_3$N nanodots administration dose from the assessment of the escape latency. In the Morris water maze test during the 5-day learning phase, the latency time for APP/PS1 mice

to find the survival platform (initially placed in the third quadrant) in the saline group underwent a very mild decrease from ~57.5 ± 1.5 to ~42.4 ± 6.9 s as shown previously[47]. Treatment with C$_3$N nanodots significantly shortened the latency time, indicating a remarkably improved learning capacity of AD mice (Fig. 3c). We also noted that treatment with 1 mg/kg/d dose obtained better therapeutic effect than that treated with 5 mg/kg/d dose, at the day 5 the latency time was ~19.2 ± 2.3 s vs. ~29.3 ± 2.9 s (Fig. 3c), suggesting that 1 mg/kg/d may be the optimal dose. The potential contribution of the swimming capability (swimming speed) to the learning effects was excluded because there was no distinct difference in the average swimming speed between two C$_3$N nanodots treated groups and the WT mice (Fig. 3d). Overall, these results demonstrated the efficacy of C$_3$N nanodots in the treatment and improving the learning capacity of AD mice, with an optimal dose of ~1 mg/kg/d. Hence, 1 mg/kg/d was used in the following in vivo experiments.

To further measure the spatial memory capability, the third quadrant residence time of mice was accumulated during 60 s swimming after retrieval of the survival platform on day 6 (Fig. 3e, f). C$_3$N nanodots-treated AD mice spent significantly more time in the third quadrant and crossed this target quadrant more often compared to control APP/PS1 mice (~15.9 ± 2.8 s vs. ~6.5 ± 2.5 s; ~7.2 ± 0.8 times vs. ~3.0 ± 1.3 times) (Fig. 3f, g, & h). In addition, the time to explore the new object among APP/PS1 mice was significantly reduced as compared to that of the WT mice (~0.7 ± 0.3 vs. ~0.4 ± 0.1). However, treatment with C$_3$N nanodots can remarkably prolong the time of APP/PS1 mice to explore the new object, which resulted in the recognition index (RI) of APP/PS1 mice (treated with C$_3$N nanodots) was remarkably improved to level comparable with WT mice (~0.7 ± 0.3 vs. ~0.7 ± 0.2) (Fig. 3i, j). These results were indicative that C$_3$N nanodots treatment could partially rescue these defects in APP/PS1 mice and may offer utility against AD.

Furthermore, the body weights among both C$_3$N nanodots treated and untreated AD mice increased steadily during the entire administration period (Supplementary Fig. 14) suggesting the higher biocompatibility of C$_3$N nanodots in animals. Moreover, the H&E staining in heart, liver, spleen, lung, and kidney tissue showed no distinct lesions (Supplementary Fig. 15). We also noted in the literature that GO-based nanomaterials have the ability to reverse the aggregation of Aβ and α-synuclein peptides[28,37]. However, it is worth pointing out that numerous studies have raised concerns about their potential long-term cytotoxicities, including inflammation reactions[48–50]. In light of these concerns, we conducted measurements of several inflammation markers after six months of treatment with C$_3$N nanodots. Remarkably, all investigated inflammation indexes, such as white blood cell count (WBC), lymphocyte count (Lymph#), monocyte count (Mon#), and granulocyte count (Gran#), fell within the normal healthy range (Supplementary Fig. 16). These findings strongly indicate that C$_3$N nanodots do not provoke severe inflammation reactions. Additionally, the biodistribution of C$_3$N nanodots suggests that the liver and kidney were the primary off-target organs of C$_3$N nanodots (Supplementary Fig. 17). Consequently, we examined liver and kidney function indicators, such as aspartate aminotransferase (AST), albumin (ALB), and urea (UREA), and found no significant differences in these function indices (Supplementary Fig. 18). This further supports the exceptional biocompatibility of C$_3$N nanodots. Taken together, these toxicological assessments collectively suggest that C$_3$N nanodots exhibit minimal toxicity in vivo.

Moreover, we performed the investigation of the excretion pathways of C$_3$N nanodots and discovered that urination and defecation played vital roles in their elimination from the body (Supplementary Fig. 19). On the other hand, degradation studies conducted under simulated physiological conditions, including an acidic environment similar to lysosomes and the presence of catalase with physiological concentrations of H$_2$O$_2$, revealed the degradability of C$_3$N

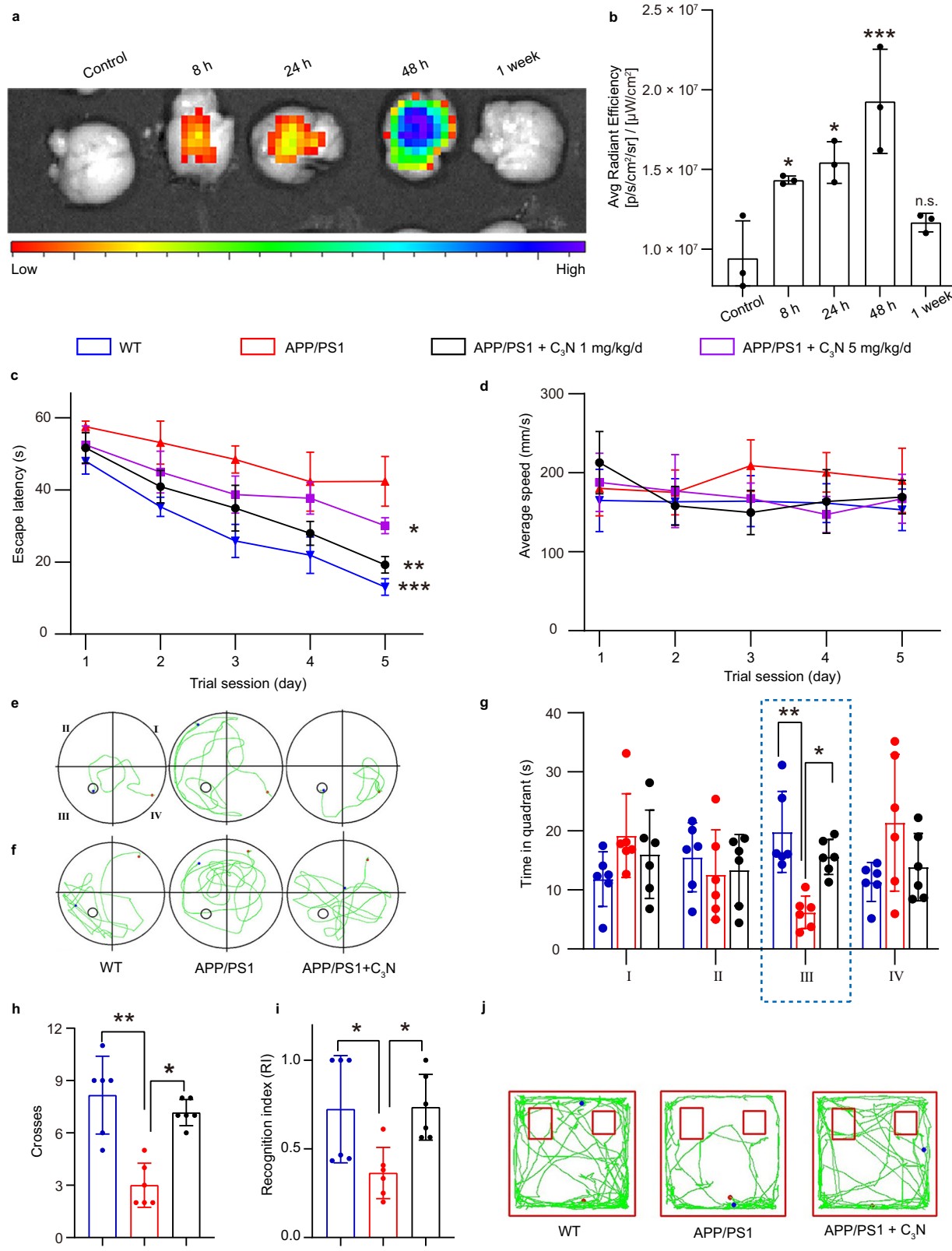

nanodots (Supplementary Figs. 20 and 21). Cell colocalization experiments further confirmed the entry of C₃N nanodots into lysosomes (Supplementary Fig. 21), implying their potential decomposition through cellular lysosome degradation mechanisms. This inherent biodegradable property may confer enhanced bioavailability and biosecurity to C₃N nanodots, highlighting their potential as a biocompatible and safe candidate.

## In vivo efficacy of C₃N nanodots against amyloid pathology

Next, we detected the level of cerebral fibrillar amyloid plaques as hallmark of AD[9] in WT and APP/PS1 mice untreated/treated with C₃N nanodots. The 6E10 anti-Aβ antibody was used because of its specific binding capability with residues 1 to 16 of the Aβ peptide. Notably, massive amyloid plaques accumulated in both the cerebral cortex and hippocampus of APP/PS1 mice treated with saline (-1.5 ± 0.5%) (Fig. 4a).

**Fig. 3 | C₃N nanodots rescues the cognition deficits of the APP/PS1 mice.**
**a**, **b** Temporal changes in fluorescence intensity of C₃N-Cy5.5 in the mouse brain following intraperitoneal injection (i.p.) at a relatively high dosage of 200 mg/kg to ensure optimal imaging. $n = 3$ mice per group. The $P$ value represents the significant difference between the C₃N nanodots-treated groups and the control group determined by one-way ANOVA ($P = 0.0331$, 0.0106, and 0.0003, respectively). **c** Time to reach hidden platform in Morris water maze of the WT and APP/PS1 mice treated without/with C₃N nanodots (Two-way ANOVA for groups, $P = 0.0396$, 0.0019, and 0.0001). **d** The average swimming velocity of each group. **e** Representative swimming paths of escape latency in the fifth day.

**f** Representative 60 s swimming paths of mice treated with various regimens to locate the escape platform after platform retrieval. **g** Accumulated time spent by mice treated with different regimens in all four quadrants. ($P = 0.0016$ and 0.0257). **h** Frequency of mice traversing the platform position after platform retrieval ($P = 0.0024$ and 0.0209). **i** The novel object recognition index (RI) of mice in each group mice ($P = 0.0311$ and 0.0168). **j** Representative paths of novel object recognition. All data are presented as mean ± SD. n.s. = no significants, $n = 6$ mice each group. Statistical significance was determined by one-way ANOVA in (**g**–**i**) with $P < 0.05$ considered statistically significant. *$P < 0.05$, **$P < 0.01$ and ***$P < 0.001$. Source data are provided as a Source data file.

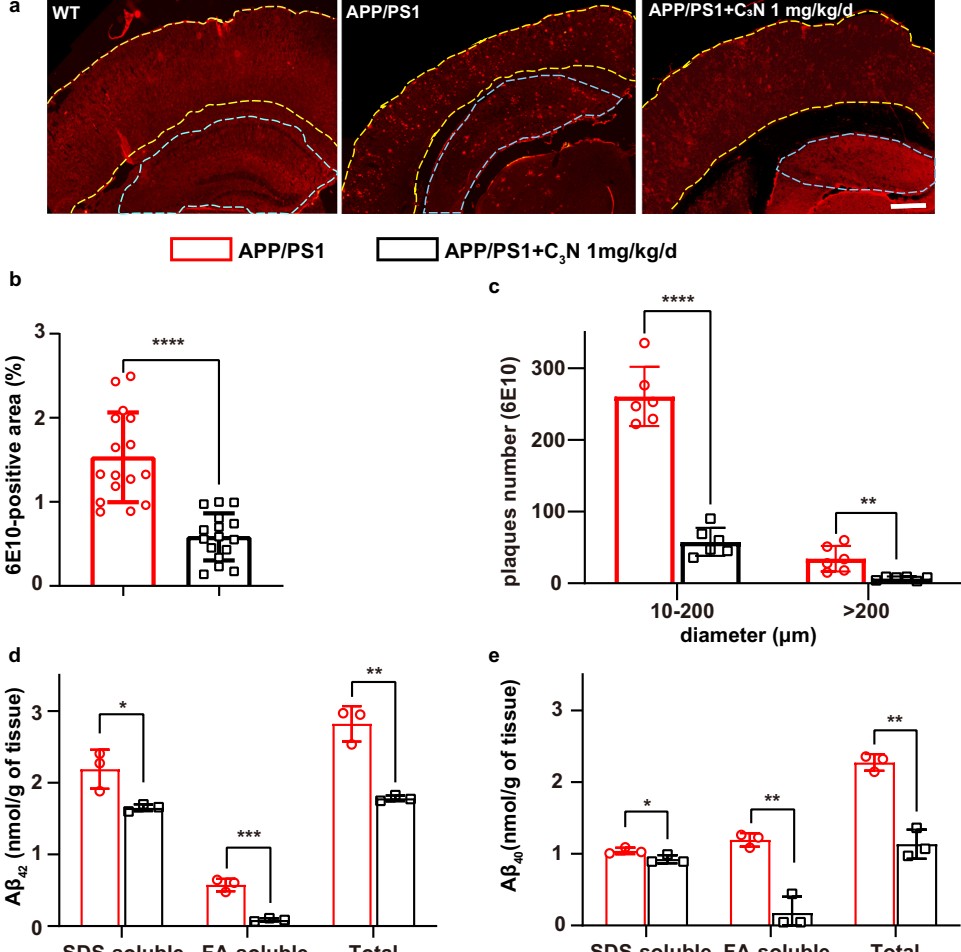

**Fig. 4 | C₃N nanodots reduce Aβ deposition levels in the brain of APP/PS1 mice.**
After six months of treatment, the whole brains of APP/PS1 mice treated with/without C₃N nanodots were collected. **a** 6E10-labeled mice brain sections immunostained for Aβ (6E10) and showing the amyloid plaque levels of the WT and APP/PS1 mice under different conditions. The cortex and hippocampus regions are marked with yellow and blue dashed lines, respectively. Scale bar = 500 μm. **b** 6E10-positive area ($n = 16$ images over 3 mice per group, $P < 0.0001$) and **c** number of 6E10-positive plaques in different sizes ($n = 6$ images over 3 mice per group,

$P < 0.0001$, $P = 0.0041$) in the APP/PS1 mice untreated/treated with C₃N nanodots at the doses of 1 mg/kg/d, respectively. **d**, **e** Levels of Aβ₄₂/Aβ₄₀ peptides in SDS–, FA–, and TBS– soluble forms in the cortex, $n = 3$ mice per group, $P = 0.0285$, 0.0007, 0.0021, 0.0498, 0.0021, and 0.0011 respectively. Statistical comparisons were performed between the APP/PS1 and C₃N nanodots-treated groups, according to the Student's $t$-test (two-tailed). Data are presented as mean ± SD.*$P < 0.05$, **$P < 0.01$, ***$P < 0.001$ and ****$P < 0.0001$ $vs$ APP/PS1 group. Source data are provided as a Source data file.

However, the amyloid plaques deposition levels remarkably decreased after treatment with 1 mg/kg/d (-0.6 ± 0.3%; a -60% decrease) C₃N nanodots treatment (Fig. 4b). These results were also confirmed by counting the number of amyloid plaques (Fig. 4c).

Considering that Aβ₄₀ and Aβ₄₂ peptides are the dominant component of the plaques in the brains of AD patients[39]. We then used enzyme-linked immunosorbent assay (ELISA) to quantify the level of intra-cephalic Aβ₄₂/Aβ₄₀ peptides. This involved using Tris-buffered saline (TBS)–, sodium dodecyl sulfate (SDS)–, and formic acid (FA)–

soluble Aβ forms corresponding to the soluble, partially soluble (non-dense plaque), and completely insoluble (dense plaque) Aβ forms, respectively. These analyses showed that treatment with C₃N nanodots decreased the level of total Aβ₄₂/Aβ₄₀ peptides by -36%/-50%, respectively. The relative FA–soluble Aβ₄₂ / Aβ₄₀ species levels was reduced most significantly by -84%/-83% (Fig. 4d, e), which suggested that C₃N nanodots effectively inhibit Aβ peptides aggregating into completely insoluble dense plaques. Overall, C₃N nanodots possessed the strong ability to delay or obstruct Aβ peptide aggregation pathogenesis in vivo.

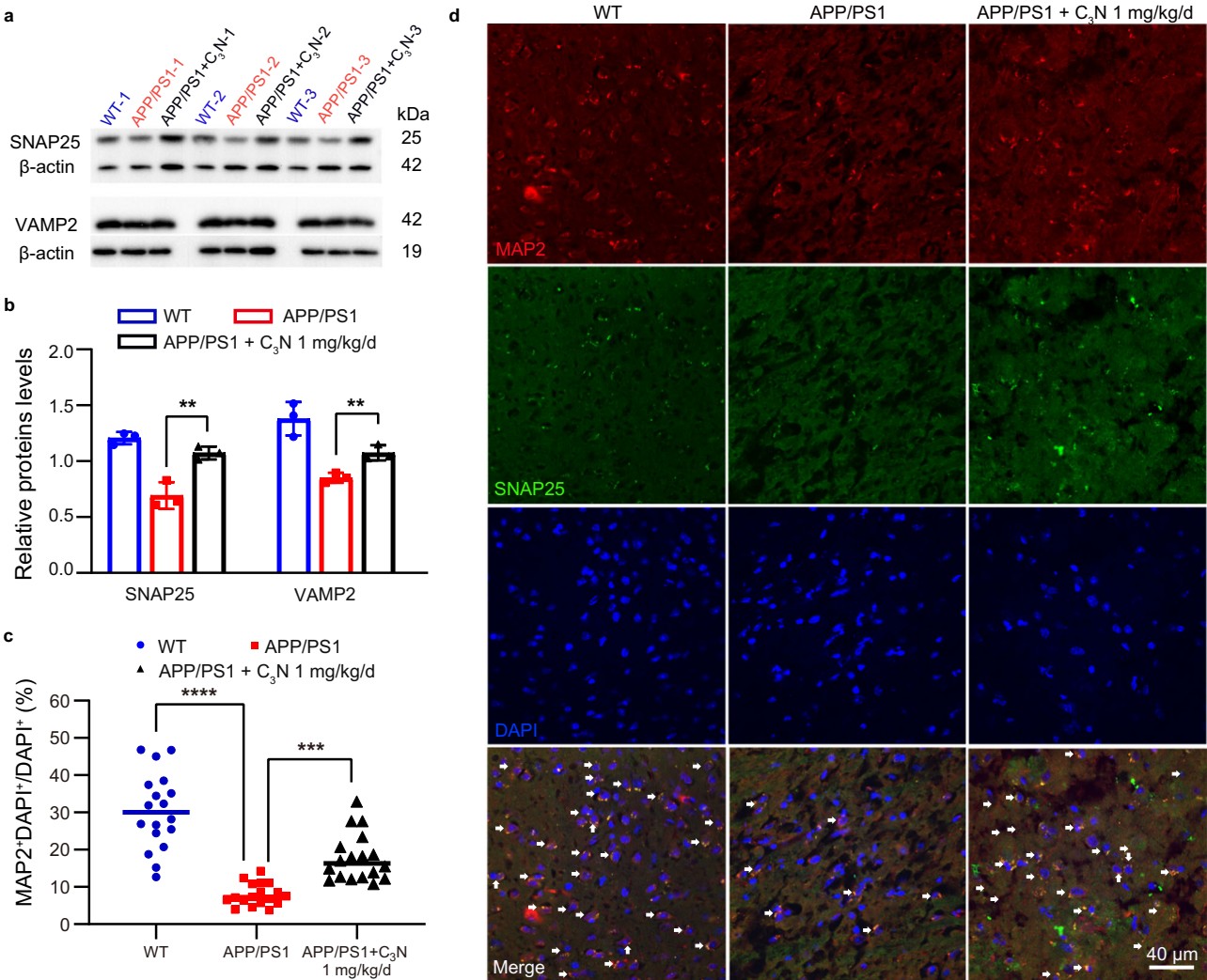

**Fig. 5 | C₃N nanodots increase the expression levels of the synaptic function-related proteins in APP/PS1 mice. a** SNAP25 and VAMP2 proteins levels were assessed using western blotting. **b** The relative expression levels of the SNAP25 and VAMP2 proteins were estimated by comparing their relative gray densities to the β-actin. $n = 3$ mice per group, Statistical significance was determined by unpaired Student's $t$ test (two-tailed) with $P < 0.05$ considered statistically significant. $P = 0.0077$ and $P = 0.0075$. **c** Quantitation of MAP2-positive neurons in cortexes. $n = 18$ micrographs examined over 3 independent mice. Statistical significance was determined by one-way ANOVA in with $P < 0.05$ considered statistically significant. $P < 0.0001$, $P = 0.0002$. **d** Immunohistochemistry on brain sections of different group mice. Representative micrographs of MAP2-labeled (red), SANP25-labeled (green) and DAPI (blue) in the cortex. Micrographs from three independent mice with similar results. All experiments were repeated three times. Data are presented as mean ± SD. **$P < 0.01$, ***$P < 0.001$ and ****$P < 0.0001$ *vs* APP/PS1 group Source data are provided as a Source data file.

## C₃N nanodots improve the level of synaptic function-related proteins in vivo

Synaptic dysfunction is another important pathological feature of AD[51,52] having a strong impact in nerves development and neurotransmitters release (including dopamine and glutamate). The SNAP25 and VAMP2 proteins are the two main synaptic proteins which protects synaptic integrity[53,54]. Therefore, we assessed the changes in expression levels of these two proteins using western blot and immunohistochemistry fluorescence assays. Western blot results demonstrated that the content of two proteins was up-regulated (Fig. 5a) after treatment with C₃N nanodots. The quantification of the SNAP25 and VAMP2 protein levels was performed using gray density analyses by utilizing Image J software. The results showed that expression levels of the two proteins were increased by 43% & 22% respectively, after treatment with C₃N nanodots (Fig. 5b).

In addition, we also examined the neuron number and synaptic damage by double-staining brain tissue with an antibody against microtubule-associated protein 2 (MAP2; a neuronal marker) and SNAP25 (a synaptic marker) (n = 3/group). The co-localization of SNAP25 and MAP2 signals reflected the expression level of synaptic proteins in neurons. Remarkably, C₃N treatment preserves MAP2-positive neuron numbers in APP/PS1 mice exposed to Aβ (Fig. 5c, d), demonstrating a ~2.3-fold upregulation. In APP/PS1 mice treated with saline only, SNAP25/MAP2 co-localization yellow pixel intensity decreased remarkably, which indicated serious dysfunction of the neural network as compared to WT mice. In contrast, treatment with C₃N nanodots for six months resulted in significantly elevated SNAP25/MAP2 expression levels (Fig. 5d). These results demonstrated that C₃N nanodots maintains an effective protective function in the synapse.

## Discussion

In this study, an effective Aβ peptides aggregation nano-inhibitor called C₃N nanodots has been explored against AD. This nano-inhibitor redirects peptide self-assembly to disordered off-pathway species and disassembly mature fibrils into smaller and amorphous entities, thereby reducing aggregation-induced neuron cytotoxicity in vitro

and in vivo. Several experimental analyses including ThT fluorescence, dot blot assays and CD spectra collectively demonstrate that $C_3N$ nanodots guide $A\beta_{42}$, $A\beta pE3$, and $A\beta_{40}$ peptides self-assembly to disordered structures rather than β-sheet-rich structures. Similarly, morphological observations using AFM and TEM imaging show that after treatment with $C_3N$ nanodots these Aβ peptides form small diffused oligomeric structures, in contrast to the long and well-defined mature amyloid fibers formed in the absence of $C_3N$ nanodots. Moreover, ThT fluorescence, dot blot assay, CD spectra, AFM, and end-to-end distance results collaborate that $C_3N$ nanodots can disaggregate the preformed long, well-defined mature fibrils into smaller, amorphous species. The results from CCK-8, LDH, and Live/Dead assay reveal that $C_3N$ nanodots relieve neuron toxicity induced by Aβ aggregation and rescue neuronal death. SEM images further helped to depict that $C_3N$ nanodots protect normal neuronal morphology from Aβ aggregation-induced destruction. Furthermore, MD simulations demonstrated that both the non-specific hydrophobic and electrostatic interactions, and the specific π–π stacking and hydrogen bonding interactions between $C_3N$ nanodots and Aβ peptides synergistically obstruct the aggregation process of Aβ peptides. The inhibitory capability of $C_3N$ nanodots on peptides aggregation is notably superior to that of GRA and fullerene. This can be attributed to the polarization of the SEP of $C_3N$ nanodots, which is induced by the presence of numerous polar C–N bonds and charged edge groups (e.g., $-COO^-$ and $-NH_3^+$). These characteristics enable $C_3N$ nanodots to engage in additional electrostatic interactions with the charged residues within the amyloid peptides, distinguishing them from GRA and fullerene represented by LJ particles. Moreover, the formation of hydrogen bonds between the charged edge groups of $C_3N$ nanodots and peptides contributes to the suppression of peptides aggregation.

Fluorescence imaging experiments provide evidence that $C_3N$ nanodots are capable of effectively crossing the BBB in mice and accumulating in the brain. The cognitive abilities among the studied mice were also restored following $C_3N$ nanodots treatment. After $C_3N$ nanodots treatment, the cognitive ability of APP/PS1 mice significantly improved to levels comparable with WT mice. Several immunological experiments including immunohistochemistry fluorescence, western blot, and ELISA assays demonstrated that APP/PS1 mice experience a decrease in cerebral fibrillar amyloid plaque levels and an increase in SNAP25 and VAMP2 with $C_3N$ nanodots treatments. Furthermore, rigorous toxicological assessments, including changes in body weight, H&E staining of vital organs (heart, liver, spleen, lung, and kidney), long-term inflammation indexes, and liver and kidney function indicators, demonstrate the exceptional biocompatibility of $C_3N$ nanodots. The main excretion pathways of $C_3N$ nanodots were found to be through urination and defecation. Additionally, simulated degradation studies suggest that $C_3N$ nanodots may undergo degradation via cellular lysosome and catalase degradation pathways. This may further enhance the bioavailability and biosecurity of $C_3N$ nanodots. Conclusively, this study not only provides useful experimental and theoretical basis for the application of $C_3N$ nanodots in neuronal protection, but also offers the groundwork for subsequent optimal designs of nanomaterials targeting Aβ peptides aggregation in AD.

## Methods

### Preparation of $C_3N$ nanodots and characterizations
The synthesis of $C_3N$ nanodots was based on the method reported by our group[38]. Briefly, the aqueous solution of 2,3-Diaminophenazine (80 mL, 1.4 mM) was heated and kept at 320 °C for 36 h in a 100 mL poly (p-phenylene)-lined stainless-steel autoclave. The products were filtered by 0.02 μm alumina microporous membrane to obtain the raw $C_3N$ nanodots. Then, the raw $C_3N$ nanodots were treated with $H_2O_2$ (5 M, 80 °C for 6 h) for further oxidization. Finally, the sample was purified via membrane dialysis with the molecular weight cutoff of 50–1000 Da for 5 days, and the oxygen-modified $C_3N$ nanodots were obtained.

The transmission electron microscopy (TEM) and high-resolution TEM (HRTEM) images were obtained using transmission electron microscope with the accelerating voltage of 200 kV (Tecnai G2 F20, FEI Corporation, American). The Fourier transform infrared (FT-IR) spectra of $C_3N$ nanodots were characterized using fourier transform infrared spectrometer (Hyperion, Bruker Corporation, Germany). The UV–Vis spectra analysis utilized a UV-vis spectrophotometer (Lambda 750, PerkinElmer, American), and the X-ray photoelectron spectra (XPS) were obtained using an X-ray photoelectron spectrometer (Axis ultra DLD, Kratos, Britain).

### Preparation of Cy5.5-conjugated $C_3N$ nanodots
To achieve fluorescence labeling of the $C_3N$ nanodots, a reaction was carried out by incubating the resulting covalent $C_3N$ nanodots with the Cy5.5 monofunctional N-Hydroxysuccinimide ester (Cy5.5-NHS) in PB buffer at pH 8.0. This incubation process was allowed to proceed overnight, facilitating the successful conjugation of Cy5.5 dye to the $C_3N$ nanodots through covalent bonding. Following the reaction, any unreacted Cy5.5 dye was eliminated through the utilization of ultrafiltration. The resultant product, namely Cy5.5-conjugated $C_3N$ nanodots (denoted as $C_3N$-Cy5.5), was subsequently stored in a dark environment at a temperature of 4 °C for future applications.

### Antibodies
Primary antibodies for immunoblotting including dot blot analysis and western blot analysis were performed with antibodies against Amyloid Fibril-Conformation-Specific (mOC87, abcam, Cat#: ab201062, 1:8000), SNAP25 (Synaptic systems, Cat#: 111-002, 1:2000), VAMP2 (abcam, Cat#: ab3347, 1:1000), β-actin (4D3, Bioworld Technology, Cat#: BS6007M, 1:5000). Secondary antibodies were conjugated with peroxidase affinipure donkey anti-rabbit IgG (H + L) (#711-035-152, Jackson ImmunoResearch, 1:10,000), Peroxidase affinipure donkey anti-mouse IgG (H + L) (#715-035-151, Jackson ImmunoResearch, 1:10,000).

Primary antibodies for immunohistochemistry were directed against purified anti-β-Amyliod 1-16 (6E10, Covance, SIG-39320, 1:500); MAP2 (AP20, Millipore, Cat#: MAB3418, 1:1,000) and SNAP25 (Synaptic systems, Cat#: 111-002, 1:2,000). Secondary antibodies were conjugated with Cy™3 affinipure donkey anti-mouse IgG (H + L) (#715-165-151, 1:400), Alexa Fluor® 488 affinipure donkey anti-rabbit IgG (H + L) (#711-545-152, 1:400).

### Preparation of monomic $A\beta_{42}$ peptides
**Synthetic** $A\beta_{42}$ (NH2-DAEFRHDSGYEVHHQKLVFFAEDVGSNK-GAIIGLMVGGVVIV-COOH, purity ≥ 98%), $A\beta pE3$ (NH2-Pyr-FRHDSGYEVHHQKLVFFAEDVGSNKGAIIGLMVGGVVIV-COOH, purity ≥ 98%) and $A\beta_{40}$ (NH2-DAEFRHDSGYEVHHQKLVFFAEDVGSNK-GAIIGLMVG GVV-COOH, purity ≥ 98%) peptides were purchased from APeptide Co., Ltd (Shanghai, China) and prepared according to protocols previous described[55,56]. Briefly, $A\beta_{42}$/$A\beta_{40}$/$A\beta pE3$ peptides were first dissolved in hexafluoroisopropanol (HFIP, 10522, Sigma Aldrich) and sonicated for 10 min. The $A\beta_{42}$–HFIP solution was then incubated at room temperature for 1 h to ensure the monomerization and structural randomization of peptides, and placed into a fume hood to completely evaporate HFIP. The obtained peptide film was stored at –80 °C. Immediately before use, the peptide film was resuspended to 5 mM in dimethyl sulfoxide (DMSO, D2650, Sigma Aldrich) and diluted to a final concentration of 100 μM in phosphate-buffered solution (PBS, 0.1 M). The solution was then centrifuged at 16,000 × $g$ for 10 min at 4 °C to remove the pre-formed fibers.

In the aggregation experiment, $A\beta_{42}$ (100 μM) was mixed with $C_3N$ nanodots at various concentrations or PBS solution to a final concentration of 50 μM and then incubated at 37 °C with constant agitation at 300 rpm for 24 h.

## Preparation of Aβ₄₂ fibrils

Aβ$_{42}$ peptide stock solutions were prepared by dissolving them in DMSO and phosphate buffer (pH = 7.4) to achieve a final concentration of 200 μM. The peptides were then aggregated for 48 h at 37 °C and centrifuged at 16,000 × $g$ for 10 min to remove insoluble material. The concentration of the stock solutions was determined using the Bradford assay. Subsequently, the peptide stock solutions were diluted in 1 mM phosphate buffer (pH = 7.4) to a final concentration of 50 μM. The samples were incubated with different concentrations of C$_3$N nanodots at 37 °C with continuous shaking at 300 rpm. The ability of C$_3$N nanodots to disaggregate mature fibrils was assessed using the ThT fluorescence assay, Dot blot assay, CD spectra, and AFM images.

## Thioflavin-T (ThT) assay

Fluorescence with Thioflavin T (ThT) was used to detect aggregated Aβ containing β-sheets[57]. A 50 μL sample was mixed with 150 μL ThT (20 μM, T3516, Sigma Aldrich) in a 96-well plate. The resulting fluorescence intensity was detected immediately after mixing with a fluorescence plate reader (BioTek, USA) at excitation and emission wavelengths of 450 nm and 485 nm, respectively. Fluorescence values of C$_3$N nanodots and ThT were subtracted from that of the mixed solution. Error bars (±s.d.) of triplicate samples are shown for selected data points.

## Dot blot assay

Dot blot assays were carried out with amyloid fibril conformation specific antibody to probe the formation level of Aβ$_{42}$ amyloid mature fibers. Briefly, 5 μL aliquots of the sample were dropped onto nitrocellulose membranes (1060002, GE Healthcare). Once the membranes dried, they were blocked for 1 h with 3% nonfat milk in tris-buffered saline (TBS) solution and then incubated with Anti-Amyloid Fibril antibody (mOC87) overnight at 4 °C. The membranes were washed 3 times in TBST for 5 min and then incubated with the horseradish peroxidase (HRP)-conjugated donkey anti-rabbit secondary antibody for 2 h at room temperature (Fig. 1a and Supplementary Figs. 4b, 5b, 6b). Finally, the membranes were developed by chemiluminescence using ECL Plus (P0018S, Beyotime).

## Atomic force microscope (AFM)

Here, 10 μL of each sample was dispersed on freshly cleaved mica sheets. After air-drying, samples were scanned and analyzed using the tapping mode of AFM (Bruker, Germany), and the height of the sample was recorded.

## Transmission electron microscopy (TEM)

Ten microliters of each sample were dispersed on a copper grid (carbon and formvar coated 300 mesh, Zhongjing Technology Co., Ltd, China) for 2 min at room temperature. Then, they were washed twice with ultrapure water and negatively stained with 1% uranyl acetate for 2 min. After air-drying, images of peptides were observed using a Tecnai G2 spirit BioTwin TEM at 120 kV.

**Circular dichroism (CD) spectroscopy.** All samples were diluted six times under PBS conditions. Spectra were detected using a Jasco J-815 circular dichroism spectropolarimeter (1 mm path length cuvette) at 25 °C. The spectrum of PBS was set as the baseline. Each sample was scanned three times and the average value was adopted. Raw data, after subtracting the buffer spectra, were smoothed according to the manufacturer's instructions.

## Primary neuron cultures

Mouse primary cortical neurons were obtained from embryonic day 18 C57BL/6J mice. All animal procedures followed the policies of the Soochow University Animal Care and Use Committee (SUACUC). In brief, dissociated neurons were plated onto dishes coated with poly-D-lysine (P6407, Sigma Aldrich) then suspended in culture medium (Neurobasal Media (21103-049, Invitrogen) containing 2% B-27 (17504-044, Invitrogen), 1% penicillin/streptomycin (15140122, P/S, Gibco), 1% L-glutamine and 0.25% GlutaMax™ (35050, Invitrogen)). Next, the plating medium was substituted with feeding medium (Neurobasal medium supplemented with 2% B27, 1% P/S, and 1% L-glutamine) on the second day after cell plating. The medium was replaced twice a week and the cultures were incubated in a 5% CO$_2$ incubator at 37 °C. Cells were used for experimentation 8 days after seeding.

## Primary astrocyte cultures

Primary astrocyte cultures were extracted from the cerebral cortex of 1-3-d-old rats (Sprague-Dawley). In brief, dissociated cortical cells were suspended in DMEM media (sh30022.01b, Hyclone) containing 1% P/S (Gibco) and 10% Fetal bovine serum (10099141, Gibco) and plated on PDL-coated 75 cm² flasks at a density of 6 × 10⁵ cells/cm². Monolayers of type 1 astrocytes were harvested 12–14 days after plating. Non-astrocytic cells were separated and removed from the flasks by shaking and changing the medium. Astrocytes were dissociated through trypsinization and reseeded on uncoated 96-well plates. The cells grew to 80–90% confluence before exposure to C$_3$N nanodots.

## In vitro cytotoxicity study

The cytotoxicity of C$_3$N nanodots was assessed using a standard CCK-8 assay (CK04, Dojindo). Primary mouse neurons, rat adrenal pheochromocytoma cells (PC12, CRL-1721, purchased from ATCC), primary rat astrocytes, human umbilical vein endothelial cells (HUVCEs, PCS-100-013, purchased from ATCC), and human neuroblastoma cells (sh-sy5y, CRL-2266, purchased from ATCC) were selected for the study. Briefly, cells in the logarithmic growth phase were seeded at a density of 5 × 10³ cells per well in 96-well plates and cultured in complete DMEM medium (#11965092, Gibco) containing 10% FBS (#03.U16001DC, EallBio) and 1% penicillin/streptomycin (#15140163, Gibco) at 37 °C with 5% CO$_2$. The cells were then co-cultured with various concentrations of C$_3$N nanodots (0, 50, 100, 150, 200, 300, 400, and 500 μg/mL) in serum-free DMEM medium until they reached approximately 80% confluence. After a 24-h incubation period, the cells were washed three times with PBS. The CCK-8 assay was performed according to the manufacturer's instructions. The absorbance (optical density, OD) of cells in different groups was measured at 450 nm using a microplate reader (Bio-Tek Instruments, Synergy NEO, USA) to calculate the cell viability using the following equation:

$$\text{cell viability}(\%) = ((\text{OD}_{test} - \text{OD}_{blank})/(\text{OD}_{control} - \text{OD}_{blank})) \times 100\% \quad (1)$$

where, $\text{OD}_{test}$ refers to the absorbance of the cells exposed to the nanomaterial sample, $\text{OD}_{control}$ refers to the absorbance of the control sample, and $\text{OD}_{blank}$ refers to the absorbance of the blank well. Each sample was tested in five replicates.

To compare the cytotoxicity between C$_3$N nanodots and GO nanosheets, a standard CCK-8 assay was performed using mouse brain microvascular endothelial cells (bEnd.3, cl-0598, purchased from Procell), BV2 murine microglial cells (BV2, cl-0493, purchased from Procell), and HUVCEs. GO was purchased from TimeNano (product model: TNWGO-3; more characterizations were provided in our previous literature[58]). The cells were co-cultured with different concentrations of C$_3$N nanodots or GO nanosheets (0, 62.5, 125, 250, and 500 μg/mL) in serum-free DMEM medium for 24 h. The aforementioned standard protocol was then followed.

## Evaluation of C₃N nanodots relieve the neurotoxicity of Aβ₄₂ oligomers

Cytotoxicity of Aβ$_{42}$ oligomers on primary neuron was evaluated using CCK-8 kit, LDH cytotoxicity assay kit (K311-400, Biovision), and Live/Dead kit (l3224, Invitrogen). Before experimentation, the neuron culture medium was used to dilute 5 mM Aβ$_{42}$ peptide stock solution and

$C_3N$ nanodots solution to achieve a mixture of 50 μM $A\beta_{42}$ and $C_3N$ nanodots at various concentrations (e.g., 100, 200, 300, 400, and 500 μg/mL). A control group with medium solution and experimental groups with 50 μM $A\beta_{42}$ peptide solution and 500 μg/mL $C_3N$ nanodots solution were analyzed. The culture solutions were incubated at 4 °C for 24 h and then added to cells for another 24 h at 5% $CO_2$ humidified environment 37 °C.

The LDH assay was performed according to LDH cytotoxicity assay kit instructions. A group of cells treated with 1% Triton X-100 was added as a positive control; the cell-free group was the negative control. Optical density at 490 nm was measured on a microplate reader and the cytotoxicity of each group was calculated according to:

$$cytotoxicity(\%) = (\text{Test Sample} - \text{Negative Control}) / (\text{Positive Control} - \text{Negative Control}) \times 100\% \quad (2)$$

For the Live/Dead assay, the prepared dye was incubated with cells for 15 min according to the Live/Dead kit instructions. Cells were then photographed under a fluorescence microscope (Leica, Germany), and live vs. dead cells were counted using Image J software.

### Morphology observation of primary neuron

Primary neurons were planted on cell culture slides, washed twice with PBS, and fixed overnight with 2.5% glutaraldehyde at 4 °C for morphological observation. Twenty-four hours later, they were washed 3× with ultrapure water for 5 min each. Then, 30%, 50%, 70%, 80%, 90%, 95%, and 100% ethanol dehydration occurred in sequence for 10 min. Gold was then sprayed on the surface of the sample, and cell morphology was observed using a scanning electron microscope (SEM, Zeiss, Germany).

### Animals and drug treatment

APP/PS1 [B6C3-Tg (APPswePSEN1dE9)/Nju] double transgenic AD mice and C57BL/6J mice were used in this study (Nanjing Model Animal Research Center, Nanjing, China). All experiments were reviewed and approved by the Animal Ethics Committee of Soochow University (Nos.: SUDA201807A422 and SUDA201907A025). APP/PS1 mice were produced and maintained on a C57BL/6J hybrid background with free access to chow and drinking water under a 12-h light/dark cycle under constant temperature (22 ± 1 °C) and humidity (40–70%).

Only male mice were tested in this study. APP/PS1 mice were randomly divided into three groups. The positive control group was intraperitoneally (i.p.) injected with vehicle (saline; APP/PS1 group). The other two groups were injected intraperitoneally with either 1 mg/kg or 5 mg/kg $C_3N$ nanodots solution. Littermate WT mice treated with saline solution were used as negative controls (WT group). Drugs were given once per day from 3 months of age for six months.

### Biodistribution study

Healthy male C57BL/6J mice of 6 months old were selected for ex vivo fluorescence (FL) imaging to verify the biodistribution of C3N-Cy5.5 nanodots. The C57BL/6J mice were purchased from from the SLACCAL Lab Animal Ltd (Shanghai, China) and maintained on C57BL/6J background. The experiment was reviewed and approved by the Animal Ethics Committee of Soochow University (No.: SUDA202007A648). The C57BL/6J mice were sacrificed at 8 h, 24 h, 48 h, and 1 week after i.p. injection of $C_3N$-Cy5.5 nanodots, administered at a dosage of 200 mg/kg ($n = 3$ per group). The main organs were collected for FL imaging and semiquantitative biodistribution analysis. The distribution of $C_3N$-Cy5.5 nanodots was tracked using an IVIS Spectrum Imaging System (PerkinElmer, USA), and FL imaging was performed at specific time points after the injections. The excitation wavelength used was 680 nm, and the emission wavelength was 710 nm.

### Excretion study

The C57BL/6J mice were purchased from the SLACCAL Lab Animal Ltd (Shanghai, China) and maintained on C57BL/6J background. The experiment was reviewed and approved by the Animal Ethics Committee of Soochow University (No.: SUDA202007A648). Three healthy male C57BL/6J mice of 6 months old were intraperitoneally injected with Cy5.5-labeled $C_3N$ nanodots at a dosage of 100 mg/kg. To monitor the excretion and distribution of the nanodots, each mouse was individually placed in a metabolic cage to facilitate the collection of urine and feces at predetermined time intervals. Following each collection, the metabolic cage was thoroughly washed and disinfected to ensure cleanliness and prevent cross-contamination. The collected urine and feces samples were carefully preserved at a temperature of −80 °C until further analysis. To determine the concentration of $C_3N$-Cy5.5 nanodots in the metabolites, the samples were subjected to measurement using Cy5.5 fluorescence. The results were then expressed as the percentage of the injected dose per gram/milliliter of feces/urine, providing insights into the excretion dynamics and distribution patterns of the nanodots.

### Evaluation of intracellular biodegradation of $C_3N$ nanodots.

$C_3N$ nanodots (1 mg/mL) were dissolved in 0.3 M acetate buffer at a pH of 5.0, creating an acidic condition similar to lysosomes. The resulting solution was incubated in a shaker at 120 rpm and 37 °C. The absorbance of the solution was continuously monitored to evaluate the acid-responsiveness of the $C_3N$ nanodots. The degradation rate (R) of the $C_3N$ nanodots was determined using the following equation:

$$R = (A_0 - A_t)/A_0 \times 100\% \quad (3)$$

Where $A_0$ represents the initial absorbance value (OD#) of the solution at 0 h, and $A_t$ represents the absorbance value at time point t (t from 0–48 h).

### The biodegradation of $C_3N$ nanodots in vitro

The in vitro degradation behaviors of $C_3N$ nanodots in biomimetic microenvironments were investigated. $H_2O_2$ is typically present in the bio- microenvironment at a physiological concentration ranging from $50 \times 10^{-6}$ to $100 \times 10^{-6}$ M. Additionally, catalase, a common enzyme found in neutrophils, which are the main components of blood in the liver, was included in the study. To perform the experiment, $C_3N$ nanodots (1 mg/mL) and catalase (200 μg/mL) in 0.01 mol/L PBS (pH = 7.00) were transferred into a vial. The resulting mixture had a total volume of 20 mL and was incubated at 37 °C in the dark for 24 h. Subsequently, $H_2O_2$ (500 μmol/L) was added to initiate the biodegradation process. The sample was placed on a magnetic stirrer and subjected to constant shaking at 220 rpm. To compensate for $H_2O_2$ consumption, an additional 200 μL of $H_2O_2$ (500 μmol/L) was added each day. After 14 days of degradation, the sample was collected for transmission electron microscopy (TEM) measurement, allowing for the evaluation of structural changes and degradation effects.

### In vitro cellular uptake

The mouse brain endothelial cell line, bEnd.3 (Procell, China), was seeded at a density of $2 \times 10^5$ cells per well in glass bottom cell culture dishes (Nest, 801001). The cells were cultured in high glucose DMEM medium supplemented with 10% FBS and 1% penicillin-streptomycin at 37 °C in a 5% $CO_2$ atmosphere. After overnight incubation at 37 °C, the cells were treated with $C_3N$-Cy5.5 at a concentration of 1 mg/mL for 5 h. PBS-treated cells were used as the negative control. Subsequently, the culture medium was replaced with Hochest 33342 dye (KeyGEN DIO tech, KGA212-50) and lysosome tracker (Invitrogen, L7526), and the cells were further incubated for an additional 20 min. After washing twice with PBS, the cells were observed using a confocal microscope (Olympus, FV1300, Japan). The Hochest 33342 channel (λex = 405 nm

and λem = 460 nm), lysosome tracker channel (λex = 504 nm and λem = 511 nm), and Cy5.5 channel (λex = 640 nm and λem = 668 nm) were chosen to visualize the cell nuclei and the uptake of Cy5.5-labeled $C_3N$, respectively.

## Tissue preparations

After behavioral tests, each group of mice was subdivided into two additional groups. In the first group, mice were subjected to cardiac perfusion under deep anesthesia and perfused with PBS and 4% paraformaldehyde (PFA, 158127, Sigma Aldrich) dehydrated with sucrose. Simultaneously, the major organs of the mice were meticulously harvested at each designated time point, followed by fixation in neutral buffered formalin (10%). Subsequently, the specimens were subjected to routine processing, wherein they were embedded in paraffin and sectioned into 8 μm slices. These sections were stained utilizing the standard hematoxylin and eosin (H&E) protocol, and their examination was conducted under a microscope. In the second group, blood samples were collected from the mice through the extraction of ocular blood. To perform hematological analysis, 100 μL of the collected blood samples were carefully transferred into anticoagulant tubes, allowing for routine blood analysis. The remaining blood samples were kept at a temperature of 4 °C for a duration of 4 h. Following this, the blood samples were subjected to centrifugation, enabling the separation of blood serum, which was subsequently utilized for conducting blood biochemistry analysis. Mouse brains were harvested by decapitation, then quickly placed in –80 °C for the extraction of brain proteins.

## Western blotting analysis

Brain tissues were homogenized in cold lysis buffer (P0013C, Beyotime) containing protease inhibitor cocktail (4693116001, Roche) and centrifuged 12,000 rpm for 15 min. Supernatants were collected and the protein concentration was determined by the BCA protein assay kit (P0009, Beyotime) measured with a microplate reader. The supernatants were mixed with 5× loading buffer (#FD006, Fdbio science) incubated at 100 °C for 10 min. Each protein (15 μg) was separated by electrophoresis using a 12% SDS-PAGE gel (P0692, Beyotime) and transferred onto a PVDF membrane (ipvh00010, Millipore). The membranes were blocked by incubation with 5% non-fat milk (wt/vol) in Tris-buffered saline containing 0.1% Tween-20 (vol/vol) (TBST) for 60 min (Fig. 5a). The membranes were then incubated overnight with primary antibody (β-actin, SNAP25, VAMP2) at 4 °C. The membranes were washed thrice in TBST for 5 min and incubated with corresponding HRP–conjugated IgG secondary antibody for 2 h at room temperature (RT). The membranes were washed in TBST (3 × 5 min) before a 2-h incubation with HRP-linked secondary antibodies to rabbit or mouse accordingly at room temperature. The membranes were then visualized using chemiluminescence on ECL Plus. For the antibodies incubated in the same blots, after imaging, the blots were stripped with stripping buffer (25 mM Glycine and 1% SDS in ddH2O, pH 2.0) for 20 min at RT to remove antibodies and washed in TBST for 10 min three times. The blots were blocked at RT for 2 h in 5% non-fat milk blocking buffer in TBST and then incubated with another primary antibody. For protein quantification, densitometry was performed with ImageJ and normalized to β-actin.

## Behavioral analysis

Spatial learning and memory performance were tested using the MWM task and the novel object recognition test. The Morris water maze was conducted in a circular pool (120 cm diameter) divided into four quadrants. In the center of the third quadrant (i.e., the target quadrant), a circular platform (i.e., survival platform) with a diameter of 10 cm was placed just below the water surface (1 cm). Mice were trained four times a day for the first five days, with quadrant one as the water entry point. The time for mice to find the survival platform

within 60 s was recorded. On the sixth day, the survival platform was removed, and the time spent in each quadrant and locomotion of the mice were recorded.

For the novel object recognition test, a cube (side length of 50 cm) was used, and two identical objects (i.e., old objects) were placed symmetrically at a position 10 cm from the sidewall. Mice were placed with their backs to the objects from the perpendicular bisector of the two objects, and the exploration time of the mice was recorded for 7 min. Before placing the next mice, the chamber was cleaned with 75% ethanol. The mice were trained for three days. On the fourth day, one of the old objects was replaced with a novel object and the exploration time and path were recorded. The results are represented by the novel object recognition index (RI), which was calculated as follows:

$$RI = (\text{time to explore the new object})/(\text{time to explore the new object} + \text{time to explore the old object}) \times 100\% \quad (4)$$

Data acquisition utilized detection and analysis software of Shanghai Xinsoft Information Technology Co., Ltd.

## Immunohistochemistry

After sucrose dehydration, brain tissue was embedded with optimal cutting temperature compound (OTC, 4583, SAKURA) and sliced into 15 μm sections (CM1950, Leica, Germany). Purified anti-β-Amyloid 1-16 (6E10) was used to examine the extracellular Aβ deposits, anti-MAP2 and anti-SNAP25 were used to detect dysfunction in neuronal networks. Brian sections were stained with primary antibodies overnight at 4 °C in a humid chamber, after being washed in PBS, followed by 2 h of incubation of Cy3-conjugated or/and 488-conjugated secondary antibodies in the dark at room temperature. Fluorescent images were acquired using a fluorescence microscope (Leica, Germany) or a confocal microscope (FV1200, Olympus, Japan) following coverslipping. The number and the area of senile plaques were quantitatively analyzed by Image J software. For histopathology of major organs, the heart, liver, kidney, spleen, lung, and kidney were isolated and stained with an H&E staining kit (ab245880, Abcam).

## $A\beta_{40}/A\beta_{42}$ quantification

$A\beta_{40}/A\beta_{42}$ content was measured using enzyme-linked immunosorbent assay (ELISA). The right hemisphere was weighed and homogenized in TBS (pH 7.4, 1:12, w/v) containing a complete protease inhibitor cocktail and centrifuged. Afterward, the precipitation was centrifuged in 2% SDS and 70% formic acid. The FA-soluble fraction was neutralized with 1 M Tris (pH 11.0) and then diluted with PBS. TBS-soluble and SDS-soluble fractions were directly diluted with PBS. Quantitation was performed according to the instructions using a Human $A\beta_{40}/A\beta_{42}$ Elisa Kit (E-EL-H0542/ E-EL-M0068km, Elabscience Biotechnology). The optical density of the samples was measured with a microplate reader (BioTek, USA) at 450 nm wavelength, and the content of $A\beta_{40}/A\beta_{42}$ in the brain was calculated as moles per gram of wet tissue.

## Statistical analysis

All results are expressed as mean ± standard deviation (SD) from at least three independent experiments. The number of mice, experiments, and statistical tests are shown for each figure in the figure legend. Statistical analyses conducted using GraphPad Prism (version 9.0) and origin (version 9.0). Datasets with only two independent groups were analyzed for statistical significance using unpaired, two-tailed Student's $t$ test. Datasets with more than two groups were analyzed using one-way ANOVA. Datasets with two independent factors were analyzed using two-way ANOVA, followed by Tukey's post hoc test. All $p$ values below or equal to 0.05 were considered significant. *$P$ < 0.05, **$P$ < 0.01, ***$P$ < 0.001, ****$P$ < 0.0001.

## Simulation model system setup

The $C_3N$ used in the simulations had a diameter of ~4.5 nm corresponding to the average diameter of $C_3N$ measured in the experiments (Supplementary Fig. 7 and Supplementary data 1). The initial $A\beta_{42}$ peptide crystal structure was taken from RCSB Protein Data Bank (PDB ID: 1Z0Q)[59] (Supplementary Fig. 7). To investigate the effect of $C_3N$ on $A\beta_{42}$ aggregation, two $A\beta_{42}$ peptides were simulated in the absence or presence of $C_3N$. In the system without $C_3N$ (control system), two peptides were solvated into a 9.6 nm × 9.1 nm × 6.5 nm water box containing 17,911 water molecules. The peptides + $C_3N$ system was derived from its counterpart, by randomly adding a $C_3N$ with a minimum distance of 1.5 nm to any heavy atom of the peptide. Then, two $A\beta_{42}$ peptides + $C_3N$ were solvated into a water box (9.6 nm × 9.1 nm × 8.2 nm) containing 22,604 water molecules. $Na^+$ and $Cl^-$ ions were added to the solvent to neutralize systems and mimic the physiological conditions of 0.15 mol/L NaCl. In addition, we also compared the inhibitory effects of stacked $C_3N$ (two layers), nano graphite (simulated by two layers of stacked graphene (GRA)), and fullerene (e.g., $C_{60}$) on the aggregation of $A\beta_{42}$. The distance between two stacked $C_3N$/GRA was set at approximately 0.33 nm, while the distance between two $C_{60}$ molecules was larger than 1 nm. Four peptides were randomly placed around $C_3N$/GRA/$C_{60}$ with minimum distances larger than 1.5 nm. Subsequently, the $C_3N$/GRA/$C_{60}$ + peptides complexes were solvated in a water box with dimensions of 13.0 nm × 13.0 nm × 13.0 nm. The number of water molecules in the water box was 70,911, 70,918, and 71,274, respectively, for the $C_3N$ + peptides, GRA + peptides, and $C_{60}$ + peptides systems. For each of the three systems, two independent 300 ns production runs were conducted for subsequent analysis.

## MD simulations

The MD simulations were carried out using the GROMACS-4.6.6[60] software package with AMBER99SB-ILDN force field[61]. The VMD software was adopted to visualize the trajectories and configurations of the MD simulations[62,63]. The TIP3P water model was adopted for solvent molecules[64]. Long-range electrostatic interactions were conducted with the particle mesh Ewald method[65]. The vdW interactions were calculated with a smooth cutoff distance of 1.2 nm. Each solvated system was first minimized using the conjugate gradient method and succeeded by a 10 ns NPT relaxation at 300 K and 1 bar. During production runs, the simulation temperature and pressure were fixed at 300 K and 1 bar with the v−rescale thermostat and Parrinello−Rahman coupling scheme[66,67], respectively. A time step of 2.0 fs was used, and coordinates were collected every 20 ps. For each system, three independent 1000 ns trajectories were collected for the analysis. Periodic boundary conditions were introduced in all directions. All solute bonds were constrained at their equilibrium values by employing the LINCS algorithm[68], and water geometry was constrained with the SETTLE algorithm[69]. Electrostatic surface potential of $C_3N$ was calculated using the Adaptive Poisson-Boltzmann Solver[70,71].

## Reporting summary

Further information on research design is available in the Nature Portfolio Reporting Summary linked to this article.

# Data availability

The data that support the findings of this paper are available in the paper and supplementary information files. All the raw data are provided in a Source Data file. The PDB data-base used in the study includes PDB ID: 1Z0Q [https://doi.org/10.2210/pdb1Z0Q/pdb]. The 3D model of $C_3N$ nanodot (in pdb format) constructed in this study is provided in Supplementary Data 1. Source data are provided with this paper.

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

## Acknowledgements

We would like to thank Taimoor Khan for critical reading of the manuscript. The authors also acknowledge support from the National Key Research and Development Program of China (2021YFA1201201 and 2021YFF1200404 to R.Z.), Natural Science Foundation of Jiangsu Province (BE2022425 to Z.K.), the National MCF Energy R&D Program of China (2018YFE0306105 to Z.K.), the National Key R&D Program of China (2020YFA0406104 and 2020YFA0406101 to Z.K.), the Innovative Research Group Project of the National Natural Science Foundation of China (51821002 to Z.K.), the National Natural Science Foundation of China (U1967217 to R.Z., 22176137 to Z.Y., 52271223, 52272043, 51972216, 52202107 and 52201269 to Z.K.), the Innovative Research Group Project of the National Natural Science Foundation of China (51821002 to Z.K.), the National Independent Innovation Demonstration Zone Shanghai Zhangjiang Major Projects (ZJZX2020014 to R.Z.), the Natural Science Foundation of the Jiangsu Higher Education Institutions of China (20KJA150010 to Z.Y.), the Starry Night Science Fund at Shanghai Institute for Advanced Study of Zhejiang University (SN-ZJU-SIAS-003 to R.Z.), and BirenTech Research (BR-ZJU-SIAS-001 to R.Z.). The authors are also grateful for Carbon-based Functional Materials and Devices, and the Collaborative Innovation Center of Suzhou Nano Science & Technology, the 111 Project, and Suzhou Key Laboratory of Functional Nano & Soft Materials.

## Author contributions

R.Z., Z.Y. and Z.K. conceived and designed the research. M.Z., X.Y., X.W., and Z.K. synthesized the title material and characterization. X.Y., J.S., and S.L. carried out the molecular, cellular, and animal experiments and analyzed data. H.Z., and Z.Y. performed MD simulations and data analysis. Z.Y., X.Y., Z.K., and R.Z. co-wrote the paper. All authors discussed and commented on the manuscript.

## Competing interests

The authors declare no competing interests.
