## [Peer review file · Nature Communications]

REVIEWER COMMENTS

Reviewer #1 (Remarks to the Author):

In this study, an effective A β peptides aggregation nano-inhibitor called C3N nanodots has been explored against Alzheimers disease. The authors showed that this novel inhibitor redirects peptide self-assembly to disordered off-pathway species and reduces aggregation-induced neuron cytotoxicity in vitro and in vivo (including mouse studies). The experimental analyses including ThT fluorescence, dot blot assays and CD spectra collectively demonstrate that C3N nanodots guide A β peptides self-assembly to disordered structures rather than β -sheet-rich structures.

Given the interesting nature of this finding, its potential utility, as well as the extensive approach by both in vitro and in vivo experiments, I believe that this work is worthy of publication in nature communications.

However, there are still two fundamental questions/issues that are still not addressed and addressing them would significantly strengthen the manuscript.

1) The actual reasoning on why particularly C3N redirects AB self-assembly has not been addressed/discussed. There seems not to be a true rational behind its current design other than "it worked by good fortune".

Therefore, what makes C3N so special over existing nano materials such as fullerenes or nano graphite (stacked graphene). This is in fact the key understanding for future design. For example, the performed classical molecular dynamics simulations do not distinguish the quantum nature of the nanodot and the authors essentially simulated a common graphene/graphite flake rather than a C3N nanodot. Therefore, either C3N are not special, i.e. bucky balls and nano graphite may act similar, or perhaps it rather relates to the overall size of C3N but not to its specific nature of its surface.

2) Several nano materials used in medicine such as graphene oxide have been shown to nevertheless facilitate long term toxicity (inflammation reactions). An important question is whether C3N accumulates in the mouse' body, particularly in the brain since it passes the blood brain barrier or whether it is broken down (indicated by its metabolites).

Reviewer #2 (Remarks to the Author):

The manuscript describes the discovery of a novel amyloid-A β aggregation inhibitor for use as a therapy against Alzheimer's disease. The authors demonstrated that the so-called nanodot C3N rescues neuronal cytotoxicity, neuronal death, and neuritic damage using in vitro models. In addition, in the APP/PS1 double transgenic mouse model, the authors could show a beneficial treatment effect on amyloid plaque load, synaptic loss and memory deficits. The authors argue that nanodots C3N are biologically safe as no abnormalities were observed in a variety of mouse tissues and no adverse effects were observed on the body weight of treated mice. The mechanism of action has been studied as well using a variety of state-of-the art methodologies. It was convincingly demonstrated that the nanodots inhibited aggregation propensity of full-length Abeta 1-42, which otherwise have a high tendency to form beta-sheets fibrils. It should be noted however, that in the brain of patients with Alzheimer's disease, other amyloid-beta peptides are highly abundant like N-truncated Abeta, which very likely contribute to the molecular pathology as well. This should be at least discussed in the context of limitations of the current work.

APP/PS1 mice were treated with C3N nanodots-saline solution per day from 3 to 9 months via intraperitoneal injection. The outcome measures are generally described, but lack some important information. The Morris water maze task lacks appropriate analysis of the final day of experiments

(6th day in the present experimental setup) with the probe trial as a major readout. Besides the time in the target quadrant, the three other quadrants should be included as well (left, right, opposite). The sex of the mice should be stated as it is well known that there is a gender-related effect on pathological events in Alzheimer's disease mouse models. No treatment effect was reported on neuron numbers in the APP/PS1 mouse model. How does the daily injection of the drug in mice possibly translate into humans?

Reviewer #3 (Remarks to the Author):

The manuscript by Yin et al. presented an investigation into the usage of C3N carbon nanodots for the inhibition of Abeta amyloidogenesis. The authors used a range of in vitro and in vivo experimental techniques, combined with molecular dynamics simulations, to reach the conclusion that carbon nanodots were a potent biocompatible nanoinhibitor and hence a potential nanomedicine against Alzheimer's disease (AD).

This work, by design, content and methodologies, belonged to a growing body of literature in the field of AD nanomedicine since Linse et al. in 2007, where nanomaterials such as graphene quantum dots, lipid disks, polymers, and gold nanoparticles have been applied to retard the aggregation and toxicity of amyloid proteins such as Abeta, alpha synuclein and human islet amyloid polypeptide, in connection with AD, Parkinson's disease and type 2 diabetes. The study itself contained the typical components of nanomaterials characterisations, structural, dynamics and toxicity assays in vitro and in vivo on amyloid aggregation and toxicity, and computer simulations to further identify the mode of interaction between the nanoparticles and the amyloid protein. As such, this manuscript did not contain notable conceptual or methodological advancements but established protocols for yet another nanomaterial. The study also did not break the mold by performing new assays to elucidate the short and long-term fate and pharmacokinetics of the nanodots in vivo, or address efficacy of the IP-administered nanodots in reaching the brain. The reviewer's assessment is that this is a work more suited for a nanoscience journal.

REVIEWER COMMENTS

Reviewer #1 (Remarks to the Author):

In this study, an effective A β peptides aggregation nano-inhibitor called C₃N nanodots has been explored against Alzheimer's disease. The authors showed that this novel inhibitor redirects peptide self-assembly to disordered off-pathway species and reduces aggregation-induced neuron cytotoxicity in vitro and in vivo (including mouse studies). The experimental analyses including ThT fluorescence, dot blot assays and CD spectra collectively demonstrate that C₃N nanodots guide A β peptides self-assembly to disordered structures rather than β -sheet-rich structures.

Given the interesting nature of this finding, its potential utility, as well as the extensive approach by both in vitro and in vivo experiments, I believe that this work is worthy of publication in nature communications.

However, there are still two fundamental questions/issues that are still not addressed and addressing them would significantly strengthen the manuscript.

Author reply: We sincerely appreciate the referee's expertise and insightful comments, which are very helpful for revising and improving our manuscript.

Q1. The actual reasoning on why particularly C₃N redirects A β self-assembly has not been addressed/discussed. There seems not to be a true rational behind its current design other than "it worked by good fortune".

Therefore, what makes C₃N so special over existing nano materials such as fullerenes or nano graphite (stacked graphene). This is in fact the key understanding for future design. For example, the performed classical molecular dynamics simulations do not distinguish the quantum nature of the nanodot and the authors essentially simulated a common graphene/graphite flake rather than a C₃N nanodot. Therefore, either C₃N

are not special, i.e., bucky balls and nano graphite may act similar, or perhaps it rather relates to the overall size of C₃N but not to its specific nature of its surface.

Author reply: Thanks for the excellent comments. Indeed, there have been reports detailing the function of graphene and fullerene in inhibiting peptide aggregation (*Nanoscale* **2015**, 7, 18725; *Org. Biomol. Chem.*, **2011**, 9, 5714; *J. Nanosci. Nanotechnol.*, **2007**, 7, 1479). These materials primarily leverage their inherent hydrophobic properties to adsorb hydrophobic residues within the peptide, effectively impeding the formation of well-structured hydrophobic cores during the peptide aggregation process. Alternatively, they can directly adsorb peptides to their surfaces through hydrophobic and van der Waals (vdW) interactions. Thus, hydrophobicity plays a pivotal role in their mechanism of action. That said, it should also be emphasized that the pronounced highly hydrophobic nature of these materials poses formidable challenges for their direct application in biological systems, as they may elicit various biological toxicities, such as severe disruption of cell membrane integrity (*Nat. Nanotechnol.*, **2013**, 8, 594; *P. Natl. Acad. Sci. U.S.A.*, **2013**, 110, 12295) and denaturation of proteins (*ACS Nano* **2015**, 9, 5713; *J. Phys. Chem. C* **2011**, 115, 23323; *ACS Nano* **2015**, 9, 663). Moreover, their hydrophobicity impedes water dispersibility and consequentially reduces their bioavailability. Nevertheless, the insights gleaned from studying the mechanisms underlying peptide inhibition have motivated our interest to search for nanomedicines with more enhanced biological utility. One promising approach involves designing a drug with reduced hydrophobicity, while still maintaining a relatively planar structure that facilitates strong vdW interactions with the peptide. Introducing charged functional groups can enhance water dispersibility, and these charged groups can form specific salt-bridges with charged groups in the A β peptide, firmly anchoring the peptide to the surface of the nanomaterial and better inhibiting its sliding and conformational adjustments on the material surface. Ideally, this candidate should be synthesized using a bottom-up approach to enable better quality control. Taking these factors into consideration, we have opted to utilize C₃N nanodots as the A β peptide aggregation inhibitor and have conducted extensive large-

scale in vivo experiments to evaluate its efficacy.

As the reviewer mentioned, it is true that molecular dynamics simulations cannot distinguish the quantum properties, such as energy band structure and photoelectric properties, of C₃N from nano graphite (stacked graphene (GRA)), fullerenes (e.g., C₆₀), and other nanodots. However, molecular dynamics based on molecular force fields can partially capture the correct electronic structure-related dispersion interactions and surface electrostatic potential distribution of nanodots. These properties significantly impact the surface physicochemical characteristics of nanodots. For instance, in molecular dynamics, the hydrophobic nature of carbon-carbon nonpolar bonds in GRA and C₆₀ can be accurately reflected using only the pure Lennard-Jones potential (evidenced by the contact angle of water droplets on graphene being ~87°–~127° (*Nat. Mater.*, **2013**, 12, 866; *Langmuir* **2013**, 29, 1457; *Phys. Rev. Lett.*, **2012**, 109, 176101; *Comput. Mater. Sci.*, **2017**, 139, 216), while the contact angle between water and ideal condensed g-C₃N₄ was ~53.5° (*Appl. Surf. Sci.*, **2015**, 328, 146). Regarding C₃N, we have calculated its surface electrostatic potential distribution (**Figure S10**), which correctly represents its hydrophilic nature due to the presence of numerous polar bonds (such as C-N polar bonds and edge polar functional groups). Therefore, molecular dynamics simulations can reasonably describe some key surface properties that distinguish C₃N from GRA and C₆₀. Furthermore, it is worth noting that GRA and C₆₀ exhibit much poorer dispersion due to their hydrophobic nature, which limits their broad applications. On the other hand, C₃N is relatively hydrophilic and exhibits high water dispersity and high biocompatibility (**Figure S11, S12, S13, S14, S15, S16, and S18**).

In accordance with the reviewer's suggestion, we have compared the inhibitory effects of stacked C₃N, GRA, and C₆₀ on the aggregation of A β ₁₋₄₂. Our findings indicate that C₃N demonstrates a slightly stronger A β ₁₋₄₂ aggregation inhibition capability compared to GRA and C₆₀, as evident from the results of the secondary structure of peptides. In addition to the vdW interactions, the electrostatic interactions also contributed significantly to the adsorption of peptides onto the surface of C₃N

nanodots, in distinct difference from the cases of GRA and C₆₀. On the other hand, these charged groups (e.g., -COO⁻ and -NH₃⁺) on the edge of C₃N nanodots formed numerous salt-bridges or hydrogen bonds with charged or polar residues in A β peptides or backbones of A β peptides, thus firmly anchoring these peptides onto the surface of C₃N nanodots and significantly hindering them from forming more backbone hydrogen bonds and subsequent aggregation. We have added more discussions on this in the revised manuscript.

Figure S10. Electrostatic surface potential of C₃N was calculated using the Adaptive Poisson-Boltzmann Solver (*Proc. Natl. Acad. Sci. U. S. A.*, **2001**, 98, 10037; *Nucleic Acids Res.*, **2007**, 35, W522). Negative, neutral, and positive potentials are displayed in red, white and blue, respectively.

Figure S9. Comparison of the influence of stacked C_3N nanodots (two layers), nano graphite (stacked graphene (GRA)) and fullerenes on the aggregation of $A\beta_{1-42}$ peptides. (a) Time evolutions of the secondary structure of each residue in two $A\beta_{1-42}$ peptides in the presence of C_3N -nanodots, stacked GRA and C_{60} molecules, respectively. (b) The proportions of each structural component in the peptides. The secondary structures of residues were assigned using the DSSP definition (*Biopolymers* **1983**, *22*, 2577). (c) The final binding configurations. (d) Residual contact number between $A\beta_{1-42}$ and three nanodots. (e) The nonbonded interaction energies (including electrostatic (elec), van der Waals (vdW) interactions, and a total of them) between C_3N nanodots, stacked GRA and C_{60} molecules and peptides during the process.

Q2. Several nano materials used in medicine such as graphene oxide have been shown to nevertheless facilitate long term toxicity (inflammation reactions). An important

question is whether C₃N accumulates in the mouse' body, particularly in the brain since it passes the blood brain barrier or whether it is broken down (indicated by its metabolites).

Author reply: Thanks for the very insightful comment. Indeed, extensive literature have highlighted the diverse toxic effects induced by graphene oxide (GO) at both the cellular level and animal model, such as acute hemo-toxicity (*ACS Appl. Mater. Interfaces* **2011**, 3, 2607; *Angew. Chem., Int. Ed.*, **2013**, 52, 4986; *ACS Appl. Mater. Interfaces* **2022**, 14, 30306), cellular oxidative stress (*Toxicol. Lett.*, **2011**, 200, 201; *Biomaterials* **2013**, 34, 1562), cell membrane destruction (*Nat. Nanotechnol.*, **2013**, 8, 594; *P. Natl. Acad. Sci. U.S.A.*, **2013**, 110, 12295), and inflammation reactions (*ACS Nano* **2015**, 9, 10, 10498; *Nanoscale Res. Lett.*, **2011**, 6, 8; *Biomaterials* **2012**, 33, 6559) and so on. These effects significantly limit the bio-applications of GO. Following your suggestion, we have also thoroughly investigated whether C₃N nanodots could elicit more extensive cytotoxicity and potential inflammation reactions. Our findings, as demonstrated in **Figure S13 and S16**, reveal that C₃N nanodots exhibit much milder cytotoxicity compared to GO across various cell lines, along with excellent erythrocyte compatibility (**Figure S11**).

Figure S13. Comparative assessment of cytocompatibility between C₃N nanodots and graphene oxide (GO) nanosheets in bend.3, BV2, and HUVEC cell lines. The top panel

displays cell viability measured by CCK8 assay, while the bottom panel presents the IC_{50} values for the respective cell lines.

In addition, we conducted a comprehensive blood routine analysis of mice that received intraperitoneal injection (i.p.) of C_3N nanodots at doses of 1 mg/kg/d and 5 mg/kg/d for a duration of 6 months. The analysis focused on inflammation-related parameters, including white blood cell count (WBC), lymphocyte count (Lymph#), monocyte count (Mon#), and granulocyte count (Gran#) (**Figure S16**). Encouragingly, all these parameters remained within the normal range, indicating no significant deviations from the expected values.

Figure S16. The inflammation indexes of mice after treatment with C_3N nanodots for six months at the dose of 1 mg/kg/d. (a) White blood cell count (WBC); (b) Lymphocyte count (Lymph#); (c) Monocyte count (Mon#); and (d) Granulocyte count (Gran#).

To investigate the biodistribution of C₃N nanodots in mice, we employed Cy5.5-modified C₃N nanodots (referred to as C₃N-Cy5.5) for fluorescence imaging. Healthy C57BL/6J mice were divided into five groups: (1) a control group without treatment, and (2) groups receiving i.p. administration of C₃N-Cy5.5 nanodots for 8 hours, (3) 24 hours, (4) 48 hours, and (5) 1 week. The mice received injections of C₃N-Cy5.5 nanodots in a PBS solution at a dosage of 200 mg/kg (n = 3 per group). At the designated time points, the mice were euthanized, and major organs including blood, brain, heart, lung, liver, spleen, intestine, kidney, and muscle were collected for biodistribution analyses.

Fluorescence intensity analysis revealed a significant accumulation of C₃N nanodots in the brain, indicating that C₃N nanodots can easily penetrate BBB. The highest fluorescence intensity was observed at 48 hours post-injection, followed by a decrease to undetectable levels after one week. Besides, Liver and kidney were identified as the major off-target organs, with the highest fluorescence intensity observed at 8- or 24-hours post-injection, gradually decreasing over time.

Figure 3. (a and b) the changes in the fluorescence intensity of C₃N-Cy5.5 in the brain of mice over time after intraperitoneal injection (i.p.) at a relatively high dosage of 200 mg/kg (n = 3 per group) to ensure optimal imaging.

Figure S17. (a and b) Biodistribution of Cy5.5-modified (denoted as C_3N -Cy5.5) C_3N nanodots after intraperitoneal injection (i.p.) at the dose of 200 mg/kg at different time points.

We also investigated the excretion pathways of C_3N nanodots and found that urination and defecation played important roles in their elimination from the body (Figure S19). Additionally, degradation studies under simulated physiological conditions (e.g., acidic condition similar to lysosomes and the presence of catalase with physiological concentrations of H_2O_2) revealed that C_3N nanodots were degradable (Figure S20 and S21). Cell colocalization experiments further confirmed the entry of C_3N nanodots into lysosomes, suggesting their potential decomposition through cellular degradation mechanisms.

Figure S19. Excretion of C_3N -Cy5.5 nanodots from mice at various time points following i.p. administration at a dose of 100 mg/kg via (a) urination and (b) defecation

pathways. Error bars represent the standard deviation of the excretion data obtained from three mice per group.

Figure S20. (a and b) TEM images (top right corner, high resolution TEM image) and (c) the lateral size distribution of C₃N nanodots before/after incubation with 0.3 M acetate buffer at pH = 5.0 for 12 hours. (d) The time evolution of degradation rate of C₃N nanodots as measured by UV-vis spectra. (e) Confocal microscopic images of bend.3 cells after a 12-hour incubation with 2 mg/mL Cy5.5-labelled C₃N nanodots. C₃N-Cy5.5 nanodots exhibit a red color due to Cy5.5 (ex/em: 673/692 nm). The intracellular localization of C₃N nanodots can be clearly observed, co-located with LysoTracker Green (green channel), while Hochest33342 blue color was used for staining cell nuclei (scale bar, 20 μm).

Figure S21. (a and b) TEM images (top right corner, high resolution TEM image) and (c) the lateral size distribution of C₃N nanodots in the presence/absence of catalase and the physiological concentration (100 μM) of H₂O₂ at 37 °C for one week.

Considering the relatively high accumulation of C₃N nanodots in the liver and kidneys, we examined liver and kidney function indicators such as aspartate aminotransferase (AST), albumin (ALB), and urea (UREA) (**Figure S18**). The results showed no significant differences in these hematological indices, further supporting the exceptional biocompatibility of C₃N nanodots.

Figure S18. Some liver and kidney function indicators of mice after treatment with C₃N nanodots at the dose of 1 mg/kg/d for six months. (a) Aspartate aminotransferase (AST); (b) Albumin (ALB); and (c) Urea (UREA).

Detailed discussions on these findings have been included in the revised manuscript to provide a comprehensive understanding of the biocompatibility profile of C₃N nanodots.

Reviewer #2 (Remarks to the Author):

The manuscript describes the discovery of a novel amyloid-A β aggregation inhibitor for use as a therapy against Alzheimer's disease. The authors demonstrated that the so-called nanodot C3N rescues neuronal cytotoxicity, neuronal death, and neuritic

damage using in vitro models. In addition, in the APP/PS1 double transgenic mouse model, the authors could show a beneficial treatment effect on amyloid plaque load, synaptic loss and memory deficits. The authors argue that nanodots C₃N are biologically safe as no abnormalities were observed in a variety of mouse tissues and no adverse effects were observed on the body weight of treated mice. The mechanism of action has been studied as well using a variety of state-of-the art methodologies.

Author reply: We sincerely appreciate the referee's expertise and insightful comments, which are very helpful for revising and improving our manuscript.

Q1. It was convincingly demonstrated that the nanodots inhibited aggregation propensity of full-length A β ₁₋₄₂, which otherwise have a high tendency to form beta-sheets fibrils. It should be noted however, that in the brain of patients with Alzheimer's disease, other amyloid-beta peptides are highly abundant like N-truncated A β , which very likely contribute to the molecular pathology as well. This should be at least discussed in the context of limitations of the current work.

Author reply: We greatly appreciate your excellent comment on this important point. Indeed, the aggregation of N-truncated A β peptides (A β pE3) is believed to play a significant role in the molecular pathology of Alzheimer's disease (AD). Following your recommendation, we conducted a comprehensive study to investigate the impact of C₃N nanodots on the aggregation of A β pE3 peptides. Furthermore, we also examined the effect of C₃N nanodots on A β ₁₋₄₀ peptides, which is a major component of the plaques found in the brains of AD patients as well.

As shown in **Figure S4 and S5**, our findings demonstrate that C₃N nanodots possess the ability to inhibit or delay the aggregation of both A β pE3 and A β ₁₋₄₀ peptides. This reinforces the robust anti-aggregation function of C₃N nanodots against A β peptides. Furthermore, we observed that C₃N nanodots can also disaggregate mature fibrils of A β ₁₋₄₂ peptide (**Figure S6**). The ability of C₃N nanodots to both inhibit aggregation and disaggregate mature fibrils holds promise for their application in mitigating the

pathological processes associated with AD. These results have been included in the revised manuscript to enrich our understanding of the therapeutic potential of C₃N nanodots in AD treatment.

Figure S4. C₃N nanodots inhibit N-truncated Aβ peptides (AβpE3) fibrillization in vitro. (a) Thioflavin T (ThT) fluorescence analysis of the effect of C₃N nanodots on the aggregation of AβpE3 peptides (50 μM) after 24-hours treatment with different concentrations of C₃N nanodots. (b) Dot blot assay measuring the levels of amyloid fiber formation under different conditions using an Aβ fibrils conformation-specific antibody (mOC87 antibody) at t = 24 hours. (c) Circular dichroism (CD) spectra of AβpE3 peptides at 0 and 24 hours, without C₃N nanodots and after incubation with C₃N nanodots for 24 hours. (d) TEM images of AβpE3 peptides under various conditions after 24-hours incubation.

Figure S5. C_3N nanodots inhibit $A\beta_{40}$ peptides aggregation in vitro. (a) ThT fluorescence analysis of the effect of C_3N nanodots on the aggregation of $A\beta_{40}$ peptides (50 μ M) after 24-hours treatment with different concentrations of C_3N nanodots. (b) Dot blot assay measuring the levels of amyloid fiber formation under different conditions using mOC87 antibody, at $t = 24$ hours. (c) TEM and (d) AFM images of $A\beta_{40}$ peptides under various conditions after 24-hours incubation.

Figure S6. C_3N nanodots disaggregate the preformed $A\beta_{42}$ mature fibrils in vitro. (a) ThT fluorescence analysis of the impact of C_3N nanodots on the $A\beta_{42}$ mature fibrils after 0-, 48- and 72-hours treatment with different concentrations of C_3N nanodots. (b) Dot blot assay measuring the residual levels of preformed $A\beta_{42}$ mature fibrils after treatment with different concentrations of C_3N nanodots for 0-, 24-, 48- and 72-hours using mOC87 antibody. (c) CD spectra of $A\beta_{42}$ mature fibrils treatment with different concentrations of C_3N nanodots for 72 hours. (d) AFM images of $A\beta_{42}$ mature fibrils treatment with different concentrations of C_3N nanodots for 72 hours. (e) AFM images of $A\beta_{42}$ mature fibrils treatment with 2 mg/mL C_3N at 0-, 24-, 48- and 72-hours. (f) TEM images of $A\beta_{42}$ mature fibrils after treatment with C_3N nanodots at 2 mg/mL for 0-, 24-, 48- and 72-hours.

Q2. APP/PS1 mice were treated with C_3N nanodots-saline solution per day from 3 to 9 months via intraperitoneal injection. The outcome measures are generally described, but lack some important information. The Morris water maze task lacks appropriate

analysis of the final day of experiments (6th day in the present experimental setup) with the probe trial as a major readout. Besides the time in the target quadrant, the three other quadrants should be included as well (left, right, opposite).

Author reply: Very good point. We have added these analyses in the revised manuscript.

Figure 3e. Accumulated time spent by mice treated with different regimens in all four quadrants.

Q3. The sex of the mice should be stated as it is well known that there is a gender-related effect on pathological events in Alzheimer's disease mouse models.

Author reply: Thanks for the excellent comments. In this study only male mice were used mainly for the following three reasons: (1) hormonal stability: female mice experience hormonal fluctuations regulated by estrogen levels, which can introduce additional variability and confounding factors. Male mice have a more stable physiological cycle without periodic hormonal fluctuations; (2) consistent brain anatomy: AD primarily affects neurons and synaptic connections in the brain. Male mice have relatively consistent brain anatomy, facilitating the comparison of differences between experimental groups. Female mice, influenced by estrogen, may exhibit more variability in brain structure, leading to complex experimental results; (3)

estrogen's impact: estrogen in female mice has neuroprotective effects that could mask or alleviate AD-related pathological changes. Using male mice thus allows for a more accurate observation and evaluation of disease progression and pathological changes upon the application of nanomedicine. We have added more explanation on this in the abstract and other relevant places.

Q4. No treatment effect was reported on neuron numbers in the APP/PS1 mouse model.

Author reply: Very good point. We have added more analysis on the neuron numbers in the APP/PS1 mouse model treated with different regimens. As depicted in **Figure 5c**, the analysis demonstrates the impact of the treatments on neuron numbers, providing quantitative evidence to support our findings.

Figure 5. (c) Quantitation of MAP2-positive neurons in cortexes.

Q5. How does the daily injection of the drug in mice possibly translate into humans?

Author reply: Very good point. As noted by the reviewer, translating the initial dose in pattern animals for first-in-human (FIH) trials is a crucial step in the clinical development of therapeutic molecules that have shown promising effects in

preclinical studies. The US Food and Drug Administration (FDA) recommends a dose-by-factor approach in which the maximum recommended starting dose (MRSD) for clinical investigation in healthy human subjects is estimated by adjusting the no observed adverse effect level (NOAEL) of the new drug or biological therapeutic using allometric factors [US FDA, 2005, <https://www.fda.gov/downloads/drugs/guidances/ucm078932.pdf>], which assumes that normalizing the dose-to-body surface area will yield equivalent biological effects (*Brit. J. Pharmacol.*, **2009**, 157, 907). This estimation involves five steps: determining NOAELs in animal species, converting NOAEL to human equivalent dose (HED), selecting the appropriate species, applying a safety factor, and considering the potential adverse effects.

The general equation for determining HED is as follows (*Drug Dev. Res.*, **2018**, 79, 373):

$$\begin{aligned}
 HED \text{ (mg/kg)} & \\
 &= \text{Animal NOAEL (mg/kg)} \\
 &\times [Weight_{animal}(kg)/Weight_{human}(kg)]^{(1-b)}
 \end{aligned}$$

where the exponent "b" for body surface area is 0.67. It accounts for the difference in metabolic rate, facilitating the conversion of doses between animals and humans. Hence, for functional C₃N nanodots with a NOAEL of 1 mg/kg in a mouse weighing 0.02 kg, the HED can be calculated as:

$$HED \text{ (mg/kg)} = 1 \text{ (mg/kg)} \times [0.02 \text{ (kg)}/60(kg)]^{(0.33)} = 0.07 \text{ mg/kg}$$

Thus, for a 60 kg human, the equivalent dose is estimated to be **4.27 mg**. This HED value is usually further divided by a factor value of 10 (to increase safety of FIH); thus, the initial dose in entry into man studies might start with 0.427 mg. It should be noted that estimating the appropriate starting dose in humans and translating it across species for clinical purposes is a complex procedure. This translated dose serves only as a reference, and the optimal clinical dose must be rigorously determined through strict protocols, as no animal species can fully mimic humans in all aspects.

Reviewer #3 (Remarks to the Author):

The manuscript by Yin et al. presented an investigation into the usage of C3N carbon nanodots for the inhibition of Abeta amyloidogenesis. The authors used a range of in vitro and in vivo experimental techniques, combined with molecular dynamics simulations, to reach the conclusion that carbon nanodots were a potent biocompatible nanoinhibitor and hence a potential nanomedicine against Alzheimer's disease (AD).

This work, by design, content and methodologies, belonged to a growing body of literature in the field of AD nanomedicine since Linse et al. in 2007, where nanomaterials such as graphene quantum dots, lipid disks, polymers, and gold nanoparticles have been applied to retard the aggregation and toxicity of amyloid proteins such as Abeta, alpha synuclein and human islet amyloid polypeptide, in connection with AD, Parkinson's disease and type 2 diabetes. The study itself contained the typical components of nanomaterials characterisations, structural, dynamics and toxicity assays in vitro and in vivo on amyloid aggregation and toxicity, and computer simulations to further identify the mode of interaction between the nanoparticles and the amyloid protein. As such, this manuscript did not contain notable conceptual or methodological advancements but established protocols for yet another nanomaterial. The study also did not break the mold by performing new assays to elucidate the short and long-term fate and pharmacokinetics of the nanodots in vivo, or address efficacy of the IP-administered nanodots in reaching the brain. The reviewer's assessment is that this is a work more suited for a nanoscience journal.

Author reply: We would like to extend our sincere appreciation to the referee for their invaluable comments, which inspired us to look deeper the novelty of our current study during the revision of this manuscript.

Indeed, the 2007 Linse et al. paper (*Proc. Natl. Acad. Sci. U.S.A.*, **2007**, 104, 8691) did a fantastic job in reporting various nanomaterials in retarding the aggregation and

toxicity of amyloid proteins associated with diverse protein conformational disorders, including Alzheimer's disease (AD), Parkinson's disease (PD), type-II diabetes, and Huntington's disease. Over the course of recent years, more additional nanomaterials have emerged as promising candidates capable of inhibiting these peptide aggregations (*Nat. Nanotechnol.*, **2018**, 13, 812; *Mater. Adv.*, **2021**, 2, 1139; *Proc. Natl. Acad. Sci. U.S.A.*, **2017**, 114, E1009; *Nat. Commun.*, **2022**, 13, 1040). Several of these NPs have also demonstrated the potential to alleviate aggregation-induced cytotoxicity in cells.

However, it is worth noting that the translation of these promising results from cellular studies to animal models has been significantly lagged. Only few NPs have been reported to exhibit efficacy in animal models, with GQDs being documented as capable of ameliorating PD symptoms in mice (*Nat. Nanotechnol.*, **2018**, 13, 812). In addition, the biocompatibility of these NPs also remains largely illusive, particularly their short- and long-term cytotoxicity. An ideal candidate for AD treatment should possess a high capacity to inhibit amyloid peptide aggregation and/or disaggregate mature fibrils, and meanwhile it should be biodegradable and exhibit excellent biocompatibility with minimal short- and long-term cytotoxicity. Furthermore, it must be able to penetrate BBB, thus reducing global cerebral A β peptide levels (particularly in fibrillar amyloid plaques), and restoring synaptic loss in AD mice. Such a candidate should also contribute to the amelioration of behavioral deficits observed in APP/PS1 double transgenic AD mice. Finally, such a candidate should be relatively easy to synthesize with a superior quality control. Taken together, the screening process for identifying such a candidate with superior multifunctionality remains an immense challenge. Within this context, we present a *new highly effective candidate* in this study.

We think our current investigation offers *several novel aspects* as also noticed by both Referee #1 and Referee #2. The first one is that the anti-A β peptide aggregation mechanism of C₃N nanodots is distinct from that of GRA and C₆₀, which primarily rely on hydrophobic interactions (see Response 1 to Referee #1). Second, we have also

studied the *N-truncated A β peptides* (A β pE3), a component playing a significant role in the molecular pathology of Alzheimer's disease, to investigate the influence of C₃N nanodots. Third, the capability of C₃N nanodots in disassembling mature A β fibrils, the accumulation of C₃N in the brain, the biodistribution of C₃N in other important organs, the excretion pathways of C₃N nanodots, the biodegradability of C₃N nanodots, the long-term inflammation reactions, and the liver and kidney function of mice are all systematically studied now (partly to respond to important questions raised by both Referee #1 and #2 – hope these additional data will also make the current referee more satisfactory). These new contents should also partially resolve the referee's concern on “the short and long-term fate and pharmacokinetics of the nanodots in vivo” and “efficacy of the IP-administered nanodots in reaching the brain”. Thus, we believe that our current findings provide a fresh perspective on the potential application of C₃N nanodots in combating AD and offer a novel avenue for this grand challenge.

REVIEWERS' COMMENTS

Reviewer #1 (Remarks to the Author):

The authors addressed most of the earlier comments.

- Both the abstract and introduction assume the amyloid hypothesis to be true, i.e. that the aggregates cause disease. Recent understandings of the role of oligomers are missing.

- 2nd paragraph intro: It is not true that there is only one drug. Lecanemab got recently approved (Biogen/Eicosan) and yet another drug candidate is Donanemab.

- A few sentences down, when talking about nanomaterials, fundamental works on nano-sized objects, and the role of nanoparticle curvature and size are not cited. For example, for an overview of these works:

https://scholar.google.com/scholar?hl=en&as_sdt=0%2C5&as_vis=1&q=amyloid+aggregation+nanoparticles+size&btnG=

- Figure 1b: Does not show full inhibition.

- Figure 2: This is most likely because peptides bound to particle surface, reducing concentration in solution. This should be mentioned. The ratio of particle to peptide should be noted to see whether it is physiologically viable. Different peptide concentrations are needed without particles as a reference.

Reviewer #2 (Remarks to the Author):

The authors have appropriately addressed the questions and comments in their revised version of the manuscript.

REVIEWERS' COMMENTS

Reviewer #1 (Remarks to the Author):

The authors addressed most of the earlier comments.

Author reply: We sincerely appreciate the referee's expertise and insightful comments, which are very helpful for revising and improving our manuscript.

- Both the abstract and introduction assume the amyloid hypothesis to be true, i.e. that the aggregates cause disease. Recent understandings of the role of oligomers are missing.

Author reply: Thanks for the excellent comment. We have modified the manuscript to better represent the recent advancements on the pathogenic mechanism of AD, including the role of oligomers and Tau protein.

- 2nd paragraph intro: It is not true that there is only one drug. Lecanemab got recently approved (Biogen/Eicosan) and yet another drug candidate is Donanemab.

Author reply: Thanks for pointing us to these new monoclonal antibodies. We have now revised the manuscript to update this accurate information.

- A few sentences down, when talking about nanomaterials, fundamental works on nano-sized objects, and the role of nanoparticle curvature and size are not cited. For example, for an overview of these works:

https://scholar.google.com/scholar?hl=en&as_sdt=0%2C5&as_vis=1&q=amyloid+aggregation+nanoparticles+size&btnG=

Author reply: Thanks for pointing us those very important literature. We have included more comments in the main text along with citations to these important papers.

Figure 1b: Does not show full inhibition.

Author reply: The referee is absolutely correct. Indeed, the C₃N nanodots do not achieve complete inhibition of peptide aggregation even though they have shown great promises. We have made it clearer by explicitly stating this point in the revised manuscript.

Figure 2: This is most likely because peptides bound to particle surface, reducing concentration in solution. This should be mentioned. The ratio of particle to peptide should be noted to see whether it is physiologically viable. Different peptide concentrations are needed without particles as a reference.

Author reply: Thank you for your valuable comments. We acknowledge your observation regarding the adsorption of peptides onto the surface of C₃N nanodots, resulting in a reduction of peptide concentration in the solution. We have taken this into consideration and have expanded our discussions on this aspect in the revised manuscript. Regarding the ratio of particles to peptides in our cellular-level experiment, it is worth noting that our choice of a slightly elevated ratio was motivated by the need to expedite our research process in the context of the prolonged progression of AD. This strategy enabled us to attain the desired results in a shorter period of time. Additionally, it is important to highlight that the drug concentration applied at the cellular level is relatively high. As we transition to the animal level, achieving higher drug concentrations necessitates overcoming the blood-brain barrier, which can be accomplished through sustained and long-term administration.

The concentration of peptides used in Figure 2 was 50 μM, and such a concentration is consistent with those previously used in various molecular biology experiments. Thus, we thought a concentration of 50 μM would be sufficient to illustrate the role of C₃N nanodots in relieving aggregation-induced neuro-cytotoxicity. We do appreciate your thoughtful feedback on this and have incorporated these considerations into our revised manuscript.

Reviewer #2 (Remarks to the Author):

The authors have appropriately addressed the questions and comments in their revised version of the manuscript.

Author reply: We sincerely appreciate the referee's expertise and insightful comments, which are very helpful for revising and improving our manuscript.